# Automatically annotated motion tracking identifies a distinct social behavioral profile following chronic social defeat stress

Joeri Bordes [1,6], Lucas Miranda [2,3,6], Maya Reinhardt [1], Sowmya Narayan[1,3], Jakob Hartmann [4], Emily L. Newman [4], Lea Maria Brix[1,3], Lotte van Doeselaar [1,3], Clara Engelhardt[1], Larissa Dillmann[1], Shiladitya Mitra[1], Kerry J. Ressler [4], Benno Pütz[2], Felix Agakov[5], Bertram Müller-Myhsok [2] ✉ & Mathias V. Schmidt [1] ✉

Severe stress exposure increases the risk of stress-related disorders such as major depressive disorder (MDD). An essential characteristic of MDD is the impairment of social functioning and lack of social motivation. Chronic social defeat stress is an established animal model for MDD research, which induces a cascade of physiological and behavioral changes. Current markerless pose estimation tools allow for more complex and naturalistic behavioral tests. Here, we introduce the open-source tool DeepOF to investigate the individual and social behavioral profile in mice by providing supervised and unsupervised pipelines using DeepLabCut-annotated pose estimation data. Applying this tool to chronic social defeat in male mice, the DeepOF supervised and unsupervised pipelines detect a distinct stress-induced social behavioral pattern, which was particularly observed at the beginning of a novel social encounter and fades with time due to habituation. In addition, while the classical social avoidance task does identify the stress-induced social behavioral differences, both DeepOF behavioral pipelines provide a clearer and more detailed profile. Moreover, DeepOF aims to facilitate reproducibility and unification of behavioral classification by providing an open-source tool, which can advance the study of rodent individual and social behavior, thereby enabling biological insights and, for example, subsequent drug development for psychiatric disorders.

Stress is an essential aspect of our daily lives, which contributes to our mood and motivation. However, exposure to severe stress can have negative consequences and has become an increasing burden on society. In particular, stress-related disorders, such as major depressive disorder (MDD), have been steadily on the rise for the last decade[1]. Our understanding of the behavioral and neurobiological mechanisms related to MDD is limited, which is part of the reason for the only moderate success of current drug treatments[2]. MDD is a complex and heterogeneous disorder, and its classification is dependent on a widespread set of symptoms. An important characteristic of MDD is the impairment of social functioning and lack of social motivation, which can lead to social withdrawal from society in extreme cases[3]. In

[1]Research Group Neurobiology of Stress Resilience, Max Planck Institute of Psychiatry, 80804 Munich, Germany. [2]Research Group Statistical Genetics, Max Planck Institute of Psychiatry, 80804 Munich, Germany. [3]International Max Planck Research School for Translational Psychiatry (IMPRS-TP), 80804 Munich, Germany. [4]Department of Psychiatry, Harvard Medical School, McLean Hospital, Belmont, MA 02478, USA. [5]Pharmatics Limited, Edinburgh EH16 4UX Scotland, UK. [6]These authors contributed equally: Joeri Bordes, Lucas Miranda. ✉e-mail: bmm@psych.mpg.de; mschmidt@psych.mpg.de

addition, disturbances in social behavior are an important risk factor for developing MDD, as poor social networks are linked to lowered mental and physical health[4,5]. The impact of social interactions was highlighted during the COVID-19 pandemic, where a substantial part of society experienced little to no social interactions for a sustained period. An increasing number of studies are now reporting the enormous impact of the pandemic, emphasizing a dramatic increase in the prevalence of stress-related disorders, in particular MDD[6,7]. Unfortunately, there is still a lack of awareness of the importance of social interactions and their role in stress-related disorders. Therefore, it is crucial to increase the understanding of the biological and psychological mechanisms behind MDD, and the influence of social behavior on the development of MDD.

Along these lines, animal models have an important role in MDD research. Although unable to recreate the exact nature of the disorder in humans, they provide a controlled environment where symptoms of MDD can be investigated[8,9]. The well-established chronic social defeat stress (CSDS) paradigm is continuously used for studying symptoms of MDD in animals[10,11]. In the CSDS model, mice are subjected daily to severe physical and non-physical stressors from aggressive mice for several weeks, which results in the chronic activation of the physiological stress response system, leading to bodyweight differences, enlarged adrenals, and elevated levels of corticosterone[12]. In addition, animals subjected to CSDS show stress-related behaviors such as social avoidance, anhedonia, reduced goal-directed motivation, and anxiety-like behavior[10,13–16]. Especially CSDS-induced social avoidance behavior, which is the avoidance of a novel conspecific, is a recognized phenomenon that is used to investigate the social neurobiological mechanisms related to chronic stress exposure and stress-related disorders[11,17,18].

Currently, several social behavioral tasks can assess different constructs of social behavior, particularly the social avoidance task[18]. It is important that these behavioral tasks are conducted with control over the environment to investigate the effects of external stimuli, such as stress exposure. For decades there has been a trend to standardize and simplify these tests to allow for greater comparability and higher throughput. Unfortunately, this has led to an oversimplification of the social behavioral repertoire and increased the risk for cross-over effects by other types of behavior, such as anxiety-related behavior. Moreover, due to limitations in tracking software, the analysis of the interaction between multiple freely moving animals remained difficult, which further limited the complexity of the behavioral assessment. Social behavior is a complex behavioral construct, which relies on many different types of behavioral interactions, that often are too complicated, time-intensive, and repetitive to assess manually[19–21]. Ultimately, this can lead to poor reproducibility of the social behavioral construct, as observed for social approach behavior[22].

The current advancement in automatically annotated behavioral assessment, however, allows for high-throughput analysis using pose estimation, involving both supervised classification (intending to extract pre-defined and characterized traits) and unsupervised clustering (which aims to explore the data and extract patterns without external information)[23–28]. Importantly, the open-source tool DeepLabCut has provided a robust and easily accessible system for deep-learning-based motion tracking and markerless pose estimation[29,30]. The use of supervised classification, by defining the behavioral patterns of interest a priori, is a powerful tool that simplifies the analysis by using predefined relevant behavioral constructs without losing the complexity of social behavior. Furthermore, recent studies have shown the value of unsupervised clustering in addition to a supervised analysis, which can reveal novel and more complex structures of behavior[19,26,31–33]. By acting in a more exploratory fashion, these practices can not only assist the discovery of novel traits but also direct researchers toward the main behavioral axes of variation across cohorts of interest. In addition, both the supervised and unsupervised

analysis approaches can provide more transparency for the behavioral definition and can easily be shared via online repositories, which contributes to a more streamlined definition of behavior across different labs[21,34]. These computational tools can elevate the current understanding of the influences of stress exposure on behavior, by increasing the resolution of the observed behavioral output[35].

Therefore, the current study provides an application of our open-source tool DeepOF[36], which enables users to delve into the individual and social behavioral profiles of mice using DeepLabCut-annotated pose estimation data (Fig. 1). DeepOF provides two main workflows; a supervised behavioral analysis pipeline, which applies a set of annotators and pre-trained classifiers to detect defined individual and social traits, and an unsupervised analysis pipeline, capable of embedding the motion-tracking data of one or more animals in a latent behavioral space, pointing toward differences across experimental conditions without any label priming. Furthermore, DeepOF can retrieve unsupervised clusters of behavior that can be compared across conditions and therefore hint at previously unrecognized behavioral patterns that trigger new hypotheses. We describe a distinct social behavioral profile following CSDS in mice that can be recapitulated with both supervised and unsupervised workflows. Moreover, the current study observes a clear state of arousal upon exposure to a novel social conspecific that fades over time, which provides crucial insights for the quantification of optimal behavioral differences across time and experimental conditions.

## Results

### The supervised pipeline provided by DeepOF yields generalizable annotations

As expected, all rule-based behaviors show high performance when compared to manual labeling, which constitutes an argument in favor of simple behavioral tagging (Supplementary Fig. 1).

When evaluating the performance of the huddle classifier, balanced accuracy in the training set ($0.78 \pm 0.005$) was marginally higher than in both validation settings (suggesting no overfitting), and performance on the internal validation ($0.75 \pm 0.046$) was not significantly higher than performance on the external validation ($0.75 \pm 0.04$) suggesting excellent generalization to new datasets (independent samples t-test: $T(7.34) = -0.03$, $p = 0.51$, Supplementary Fig. 2A). In addition, pseudo-labeling conducted on the external dataset showed a strong and significant correlation between total behavior duration across manual and predicted labels (Supplementary Fig. 2B). Finally, the SHAP analysis of the deployed classifier revealed low head movement, low spine stretch, low body area, and low locomotion speed as the most important features of the model, which goes in line with the accepted definition of the behavior (Supplementary Fig. 2C).

### The physiological and behavioral hallmarks of stress are reproduced by CSDS

The CSDS paradigm was performed to maintain stress exposure for several weeks (Fig. 2A), which induced dysregulation of the hypothalamic-pituitary-adrenal axis (HPA-axis) and a stress-related behavioral profile. Male mice that were subjected to CSDS showed clear hallmarks of stress exposure, as observed by a significant increase in body weight during the stress paradigm, which was especially apparent towards the end of the stress (Fig. 2B, C), an increase in relative adrenal weight (Fig. 2D), reduced locomotion and time spent in the inner zone of the OF (Fig. 2E, F), and a significantly reduced SA-ratio in the SA task (Fig. 2G). Notably, no bodyweight difference was observed at the beginning of the CSDS paradigm (Fig. 2B).

Further exploration of the OF data using PCA across four 2.5 min consecutive time bins showed that all time bins were significantly different from each other, suggesting that they all should be included in further behavioral analysis of the OF data (Supplementary Fig. 3A,

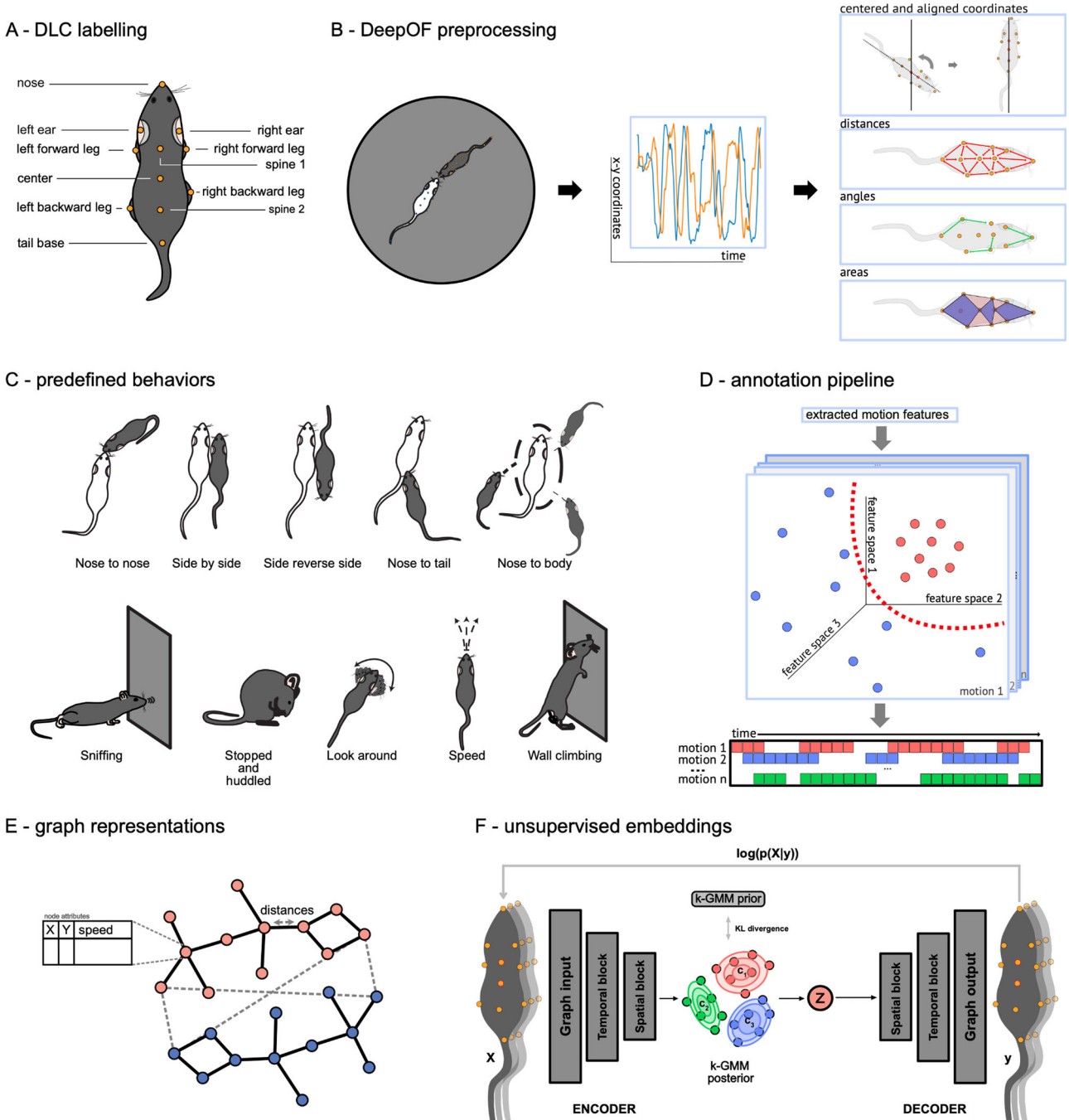

**Fig. 1 | DeepOF workflow. A** 11 labels were tagged on each annotated mouse using DeepLabCut. **B** DeepOF preprocessing pipeline. One or two mice (a C57Bl/6N experimental subject and a CD1 social companion depending on the dataset) were tagged using the provided DeepLabCut models. After tracking body parts with DeepLabCut, DeepOF was used to smooth the retrieved trajectories, interpolate outliers, and extract features (including coordinates, distances, angles, areas, speeds and accelerations). **C** Set of predefined behaviors that the DeepOF supervised pipeline can retrieve. These include dyadic motifs (such as nose-to-nose contacts) and individual motifs (such as climbing), which are reported individually for all tracked mice. The stopped-and-huddled classifier[28] is abbreviated as "huddle" in DeepOF output (not to be confused with group huddling behavior[67]). **D** Schematic representation of the supervised pipeline in DeepOF. A set of extracted motion features (only three dimensions are shown for visualization purposes) are fed to a set of rule-based annotators and pre-trained classifiers, which report the presence of each behavioral trait at each time by learning how the corresponding trait is distributed in the feature space (red dots). The set of classifiers then yields a table indicating the presence of each motif across time, which can be used for further analysis. Note that annotators are not necessarily mutually exclusive, as several predictors can be triggered at the same time. **E** Graph representation of animal trajectories used by DeepOF in the unsupervised pipeline. All 11 body parts per animal are connected using a pre-designed (but customizable) adjacency matrix. Nodes are annotated with $x$, $y$ coordinates and speed of each body part at each given time, and edges with the corresponding distances. This representation can also handle multi-animal settings, where the graphs of individual animals are connected with nose-to-nose, nose-to-tail, and tail-to-tail edges. **F** Schematic representation of the deep neural network architecture used for the unsupervised clustering of behavior. Data is embedded with a sequence-aware spatio-temporal graph encoder, and clustered at the same time by selecting the argmax of the likelihood of the components of a mixture-of-Gaussians latent posterior. Unidirectional black arrows indicate forward propagation, and gray arrows indicate the reconstruction and KL divergence terms of the loss function, the latter of which minimizes the distance to an also mixture-of-Gaussians prior.

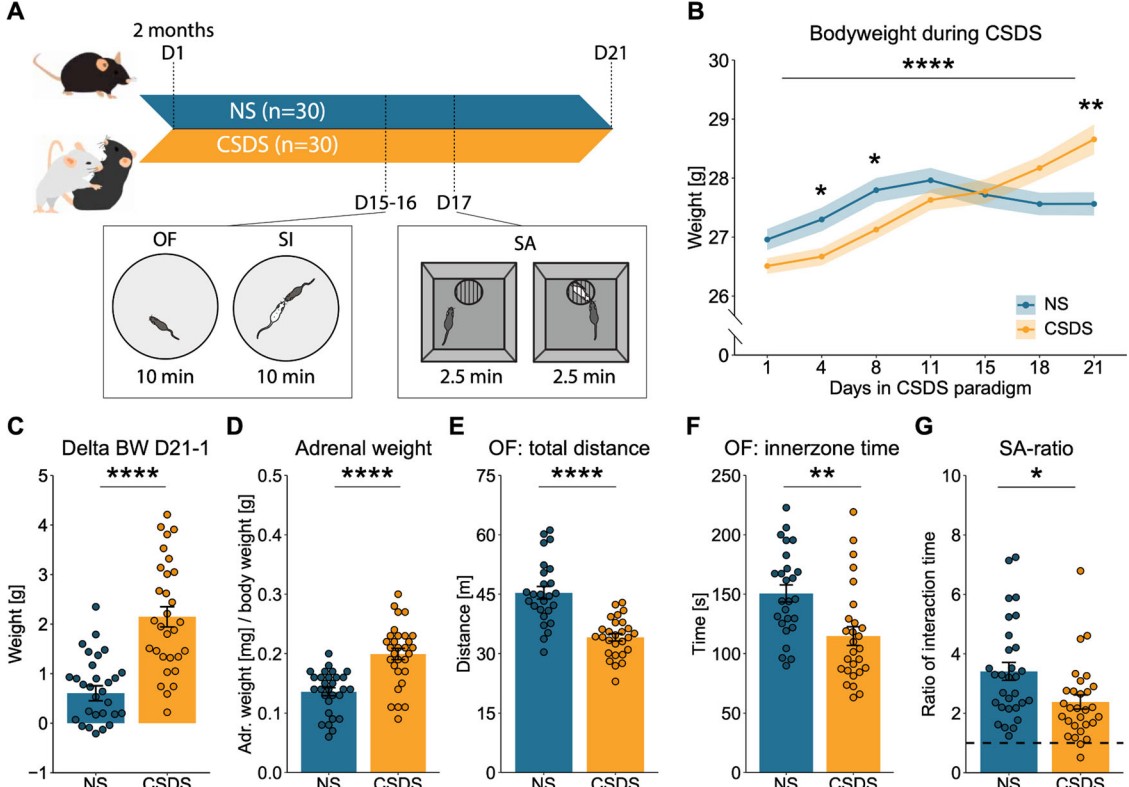

**Fig. 2 | Classical hallmarks for chronic social defeat stress. A** Experimental timeline for the CSDS paradigm and behavioral testing, including the open field (OF) and social interaction (SI) task on day 15–16 (animals were divided between the two days) and social avoidance (SA) task on day 17. **B** Significant increase of body weight after CSDS exposure (two-way ANOVA: within-subject effect of time: $F(6,406) = 13.58$, $p = 4.59e-14$, as well as time×condition interaction effect: $F(6,406) = 6.13$, $p = 3.65e-6$, but no between-subject effect on condition: $F(1,406) = 0.20$, $p = 0.653$). Post-hoc analysis with Benjamini Hochberg revealed no significant difference on day 1, 11, 15, and 18, but there was a significant difference on day 4 ($T(1,58) = 6.36$, $p = 0.033$), 8 ($T(1,58) = 6.55$, $p = 0.033$), and 21 ($T(1,58) = 11.57$, $p = 0.007$). **C** The delta body weight during the CSDS paradigm (day 21– day 1) was

significantly increased in CSDS-exposed animals (Two-tailed independent samples t-test: $T(58) = −6.09$, $p = 9.8e-8$). **D** Increase of relative adrenal weight after CSDS exposure (Two-tailed independent samples t-test: $T(57) = −5.44$, $p = 1.15e-6$). **E** The total locomotion in the OF was reduced after CSDS exposure (Two-tailed independent samples t-test: $T(51) = 6.15$, $p = 1.18e-7$). **F** The inner zone time in the OF was reduced after CSDS exposure (Two-tailed independent samples t-test: $T(51) = 3.37$, $p = 0.0015$). **G** The SA-ratio was reduced in the SA task after CSDS exposure (Two-tailed wilcoxon test: $W = 617$, $p = 0.006$). The timeline and bar graphs are presented as mean ± standard error of the mean and all individual samples as points. $N = 30$ for NS and CSDS for (**B**–**G**). Source data are provided as a Source Data file.

B). The OF PCA between conditions revealed a significant difference and showed the importance of the OF parameters, in which total distance, look-around, and sniffing came out as the top contributing behaviors (Supplementary Fig. 3C, D). A significant stress effect was observed for the total distance, look-around, and inner–zone time throughout the different time bins, whereas sniffing was altered, but not in all time bins (Supplementary Fig. 3E–J). Importantly, even though a stress-induced effect can be found in the OF task, a general habituation effect to the OF in both NS and CSDS can be observed, as total distance reduces over time, while look-around and sniffing increase. The successful habituation to the novel environment is crucial for the subsequent SI task to allow full attention to the novel social conspecific (Supplementary Fig. 3E–G).

### DeepOF social behavioral classifiers show a stronger PCA separation for stress exposure than social avoidance
The social behavioral pattern during the SI task was investigated in four non-overlapping time bins of 2.5 min each to match the time frame in the SA task. Principal component analysis (PCA) was performed to show the difference between time bins in the social behavioral profile regardless of the animal's stress condition (Fig. 3A). Interestingly, the PCA showed a significant effect between the time bins, in which the first 2.5 min time bin was significantly different from the subsequent ones (5, 7.5, and 10 min). In contrast, the subsequent time bins did not show variation between one another (Fig. 3B). This

suggests that the different time bins in the SI task are an important variable, and that the first 2.5 min time bin should be specifically investigated. Next, the SA and SI tasks were compared on their ability to distinguish between NS and CSDS animals. PCAs were performed for the SA task (Fig. 3C) and the 2.5 min time bin SI data (Fig. 3D, E), both of which showed a significant difference between the conditions in the principal component (PC) 1 eigenvalues (Fig. 3C–E). However, the SI task showed a clearer separation of the conditions than the SA task, suggesting that the SI task is a more powerful tool for identifying stressed animals than the SA task. In addition, the PC1 top contributing behaviors for the 2.5 min time bin SI data were calculated using the corresponding rotated loading scores (Fig. 3F). The top five contributing behaviors were reported as essential behaviors for identifying the stressed phenotype, which consisted of B-huddle, B-look-around, B-nose-to-tail, B-speed, and B-nose-to-body from the C57Bl/6N animal, whereas the other behaviors within the top 10 were either contributing to the CD1 animal or had a low rotated loading score (Fig. 3F). Here, "B-" indicates behaviors related to or initiated by the C57bl/6N animals, whereas "W-" refers to the CD1.

### DeepOF social behavioral classifiers are strongly altered by CSDS
Next, the influence of the CSDS on the top five contributing behaviors in the SI task was investigated. In accordance with the PCA time bin analysis, a clear stress-induced effect was observed, with elevated

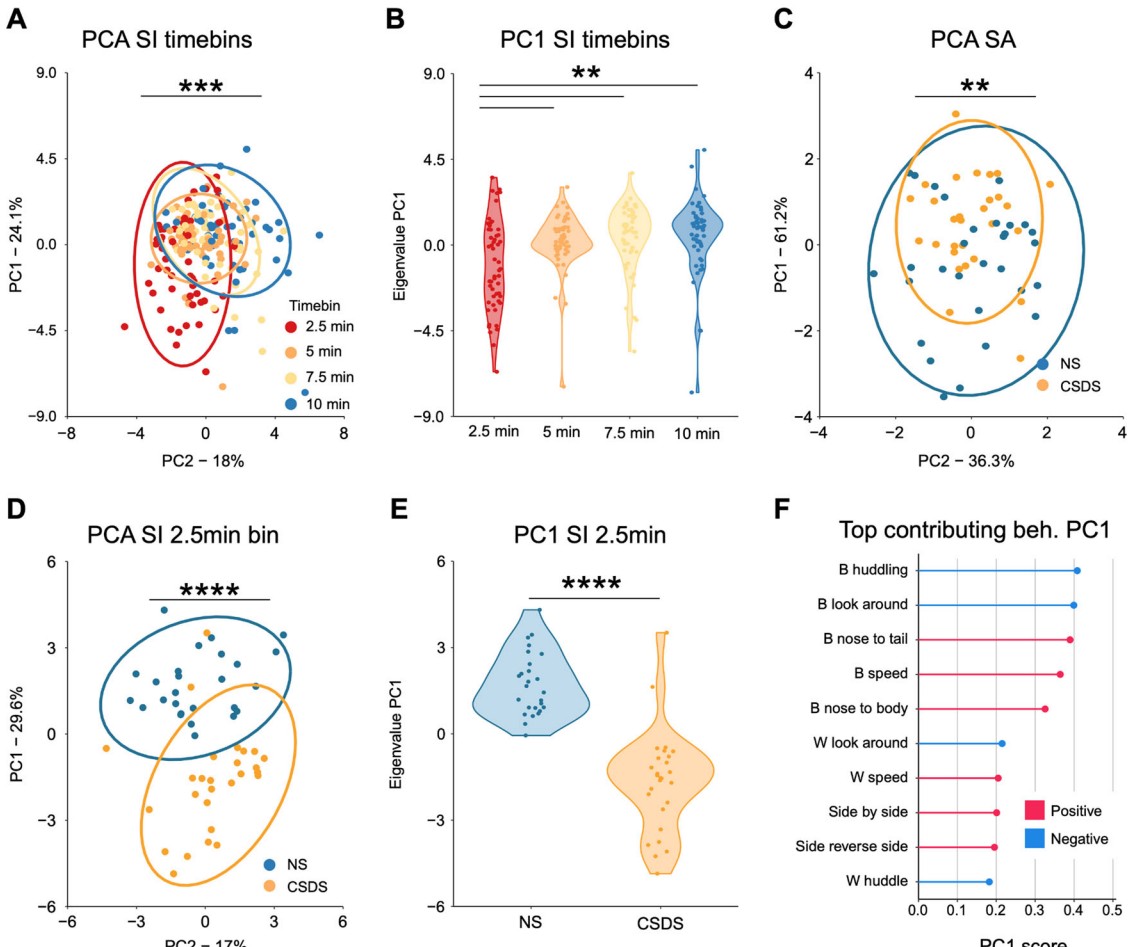

**Fig. 3 | Social interaction binning yields more separable PCA projections than the social avoidance task. A** In the SI data a PCA revealed that the first 2.5 min time bin is significantly different from the other time bins. (Kruskal-Wallis test: $H(3) = 19.90$, $p = 0.0002$. **B** The PC1 eigenvalues of the SI time bin PCA. Post-hoc Wilcoxon: 2.5 min vs. 5 min ($W = 957$, $p = 0.01$), 2.5 min vs. 7.5 min ($W = 860$, $p = 0.0018$), 2.5 min vs. 10 min ($W = 811$, $p = 0.0011$). **C** The SA task PCA showed a significant difference in the PC1 eigenvalues between conditions. The PCA data consisted of the SA-ratio, total time time spent with the non-social stimulus, and total time spent with the social stimulus. Two-tailed independent samples $t$-test: $T(57) = -2.84$, $p = 0.006$. **D** The SI 2.5 min time bin PCA showed a significant difference in the PC1 eigenvalues between conditions. The PCA data consisted of all the SI DeepOF behavioral classifiers, as listed in Fig. 1C. Two-tailed independent

samples $t$-test: $T(51) = 8.28$, $p = 5.39e-11$. **E** The PC1 eigenvalues of the 2.5 min time bin SI task. **F** The top contributing behaviors of the SI 2.5 min time bin in PC1 using the corresponding rotated loading scores. The top five behaviors were reported as the essential behaviors for identifying stress exposure (B-huddle (−0.41), B-look-around (−0.40), B-nose-to-tail (0.39), B-speed (0.36), B-nose-to-body (0.33). "B-" indicates C57Bl/6N behaviors and "W-" indicates CD1 behaviors. The PCA graphs (Fig. 3A, C, D) are provided with a 95% confidence ellipse and all individual samples as points. Further PC1 analyses (Fig. B, E) are represented with a violin plot and all individual samples as points. In Fig. 3F the absolute score of the PC1 value is represented by the point. $N = 26$ for NS and $n = 27$ for CSDS in (**A**, **B**, **D**–**F**) and $n = 30$ for NS and CSDS in (**C**). Source data are provided as a Source Data file.

duration in the CSDS animals for B-look-around (Fig. 4A, B) and B-huddle (Fig. 4C, D), while lowered for the B-speed (Fig. 4E, F), B-nose-to-tail (Fig. 4G, H), and B-nose-to-body (Fig. 4I, J). The total duration per time bin for the top contributing behaviors showed the strongest CSDS-induced effect in the 2.5 min time bin data (supplemental Fig. 4, timeline graphs), compared to the 5, 7.5, and 10 min time bins. In addition, supplemental Fig. 4 shows the 10 min total duration and time bin analyses for all other DeepOF behavioral classifiers, in which a significant stress effect is observed for B-sniffing, B-wall-climbing, and Side-by-side.

### Z-score for DeepOF social interaction correlates with Z-score for stress physiology

The Z-score of stress physiology was calculated using the relative adrenal weight and body weight on day 21 of the CSDS. The stress physiology Z-score provides a strong CSDS profiling tool and was used for correlation analysis between the SA and SI tasks. Even though the behavioral and physiological readouts were not obtained at the same

time, the former can be used as a proxy of the impact of the stress exposure, and are expected to be stable during the last week of the CSDS pipeline. No significant correlation was observed between the Z-score of stress physiology and the SA ratio (Fig. 5A). Subsequently, the Z-score of SI was calculated by using the 2.5 min time bin of the top five contributing behaviors in the SI task (Fig. 4). Stress physiology and SI Z-score showed a significant positive correlation (Fig. 5B), which indicates that the SI Z-score provides a stronger tool for CSDS profiling compared to the SA ratio. Next, correlation analyses were performed between the Z-score of SI and all other behavioral and physiological measurements which indicated a strong correlation with several OF parameters. Highly affected OF parameters, such as speed, distance, inner zone entries, and look-around might be directly related to social anxiety and warrant further investigation. Interestingly, no correlation with the SA ratio was observed (Fig. 5C).

Notably, the SA task is extensively used to distinguish resilient and susceptible animals in the CSDS paradigm[10,17], and depending on the protocol and stress severity this can give a distinction between

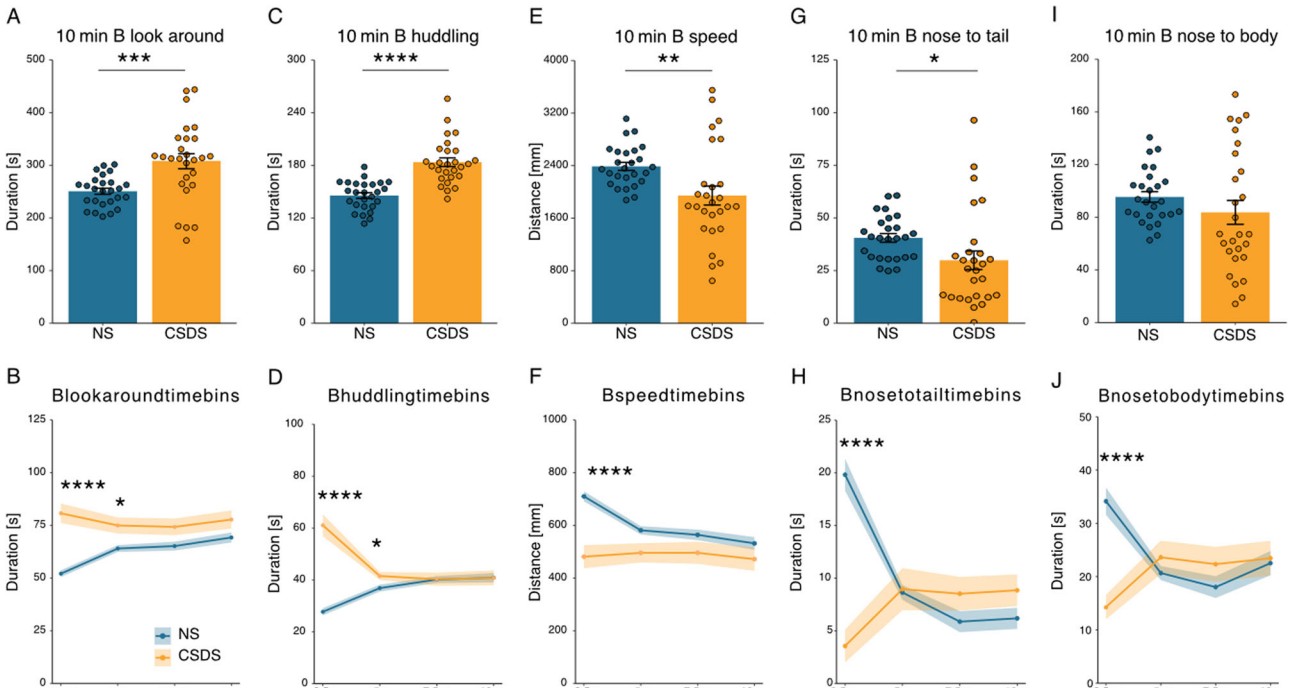

**Fig. 4 | Top contributing behaviors in the social interaction task for 10 min total duration and time bins. A** The total duration of B-look-around. Two-tailed Welch: $T(34.1) = -3.71$, $p = 0.0007$. **B** Time bin for B-look-around. Benjamini Hochberg (BH) posthoc for the 2.5 min time bin: ($T(51) = 33.46$, $p = 1.78e-6$) and the 5 min time bin ($T(51) = 6.84$, $p = 0.024$), but not for the 7.5 and 10 min time bins ($p = 0.067$, $p = 0.093$, respectively), two-way ANOVA: condition effect: $F(1,208) = 37.45$, $p = 4.59e-9$, time effect: $F(1,208) = 4.02$, $p = 0.046$, and condition × time effect: $F(1,208) = 8.87$, $p = 0.003$. **C** The total duration of B-huddle. Two-tailed independent samples *t*-test: $T(51) = -6.40$, $p = 4.8e-8$. **D** Time bin for B-huddle. Wilcoxon posthoc for the 2.5 min time bin ($W(26,27) = 63.5$, $p = 1.3e-6$), and the 5 min time bin ($W(26,27) = 204$, $p = 0.018$), but not for the 7.5- and 10 min time bins ($p = 0.52$, $p = 0.52$, respectively), Kruskal-Wallis: 2.5 min: $p = 1.25e-6$, 5 min: $p = 0.018$, 7.5 min: $p = 0.51$, and 10 min: $p = 0.51$. **E** The total duration of B-speed. Two-tailed Welch: $T(35.04) = 2.84$, $p = 0.0074$. **F** Time bin for B-speed. BH posthoc for the 2.5 min time bin ($T(51) = 22.41$, $p = 7.16e-5$), but not for the 5-, 7.5-, and 10 min time bins

($p = 0.076$, $p = 0.20$, $p = 0.24$, respectively), two-way ANOVA: condition effect: $F(1,208) = 22.60$, $p = 3.72e-6$, time effect: $F(1,208) = 7.51$, $p = 0.007$, and condition × time effect: $F(1,208) = 6.34$, $p = 0.013$). **G** The total duration of B-nose-to-tail. Two-tailed Welch: $T(36.70) = 2.18$, $p = 0.036$. **H** Time bin for B-nose-to-tail. Wilcoxon posthoc for the 2.5 min time bin ($W(26,27) = 660$, $p = 1.5e-7$), but not for the 5-, 7.5-, and 10 min time bins ($p = 0.19$, $p = 0.49$, $p = 0.49$, respectively), Kruskal-Wallis: 2.5 min: $p = 1.43e-7$, 5 min: $p = 0.18$, 7.5 min: $p = 0.48$, 10 min: $p = 0.48$. **I** The total duration of B-nose-to-body. Welch: $T(35.85) = 1.18$, $p = 0.24$. **J** Time bin for B-nose-to-body. Wilcoxon posthoc for the 2.5 min time bin ($W(26,27) = 626.5$, $p = 3.97e-6$), but not for the 5, 7.5 and 10 min time bins ($p = 0.85$, $p = 0.85$, $p = 0.85$, respectively), Kruskal-Wallis: 2.5 min: $p = 3.8e-6$, 5 min: $p = 0.85$, 7.5 min: $p = 0.85$, 10 min: $p = 0.85$. The timeline and bar graphs are presented as mean ± standard error of the mean and all individual samples as points. $N = 26$ for NS and $n = 27$ for CSDS in (**A**–**J**). Source data are provided as a Source Data file.

resilient and susceptible animals (Fig. 5D–F). Interestingly, while clearly differentiating affected and non-affected individuals, the DeepOF module does not find a distinction between SA-ratio-defined susceptibility and resiliency on the 2.5 min bin SI DeepOF behavioral classifiers (Fig. 5G–M), indicating that the DeepOF behavioral classifiers represent a unique and distinguished set of resilience-linked phenotypes.

### The DeepOF unsupervised pipeline can be flexibly applied across different experimental settings

The unsupervised pipeline within DeepOF was applied to three datasets and four settings. These included both single and multi-animal embeddings on the SI dataset, single-animal embeddings on the OF dataset, and single-animal embeddings on the SA dataset. When applying this workflow to a new dataset, the number of clusters is a hyperparameter the user must tune. In this study, an optimal solution was found by selecting the number of clusters that explains the largest difference between experimental conditions (in terms of the area under the ROC curve of a classifier to distinguish between them, see methods for details). While DeepOF could be used to describe the behavioral space of a single condition, this model selection procedure aims at maximizing the power to detect behavioral differences between experimental conditions. An optimum of 10 clusters was measured for both single- and multi-animal SI settings (Fig. 6A and

Supplementary Fig. 5A), whereas the single-animal OF setting showed an optimum of 11 clusters (Supplementary Fig. 6A), and the SA setting of 17 clusters (Supplementary Fig. 7A). Timepoint UMAP projections of the latent space depicting all clusters can be found in Fig. 6B, and Supplementary Figs. 5B, 6B, and 7B for all four settings, respectively.

### DeepOF can quantify behavioral differences over time in an unsupervised way

Once the number of clusters was fixed, the stress-induced phenotype was investigated over time in both SI and OF settings. SA was excluded of this analysis due to the shorter length of the videos (2.5 min), in which no decay of arousal should be observed in the animals. To this end, a growing time window spanning an increasing number of sequential seconds was analyzed. For each analysis, the discriminability between conditions was tested by evaluating the performance of a linear classifier to distinguish between them in the global animal embedding space, for which each experiment is represented by a vector containing the time spent per cluster (see methods for details). The bin size for which discriminability was maximized was then selected as optimal and used for further analysis. In this case, we observed an optimum of 126 and 124 s for the single-animal and multi-animal SI tasks respectively, indicating that differences between conditions are maximized early in the 10-min-long experiments, which is compatible with habituation. Furthermore, performance across

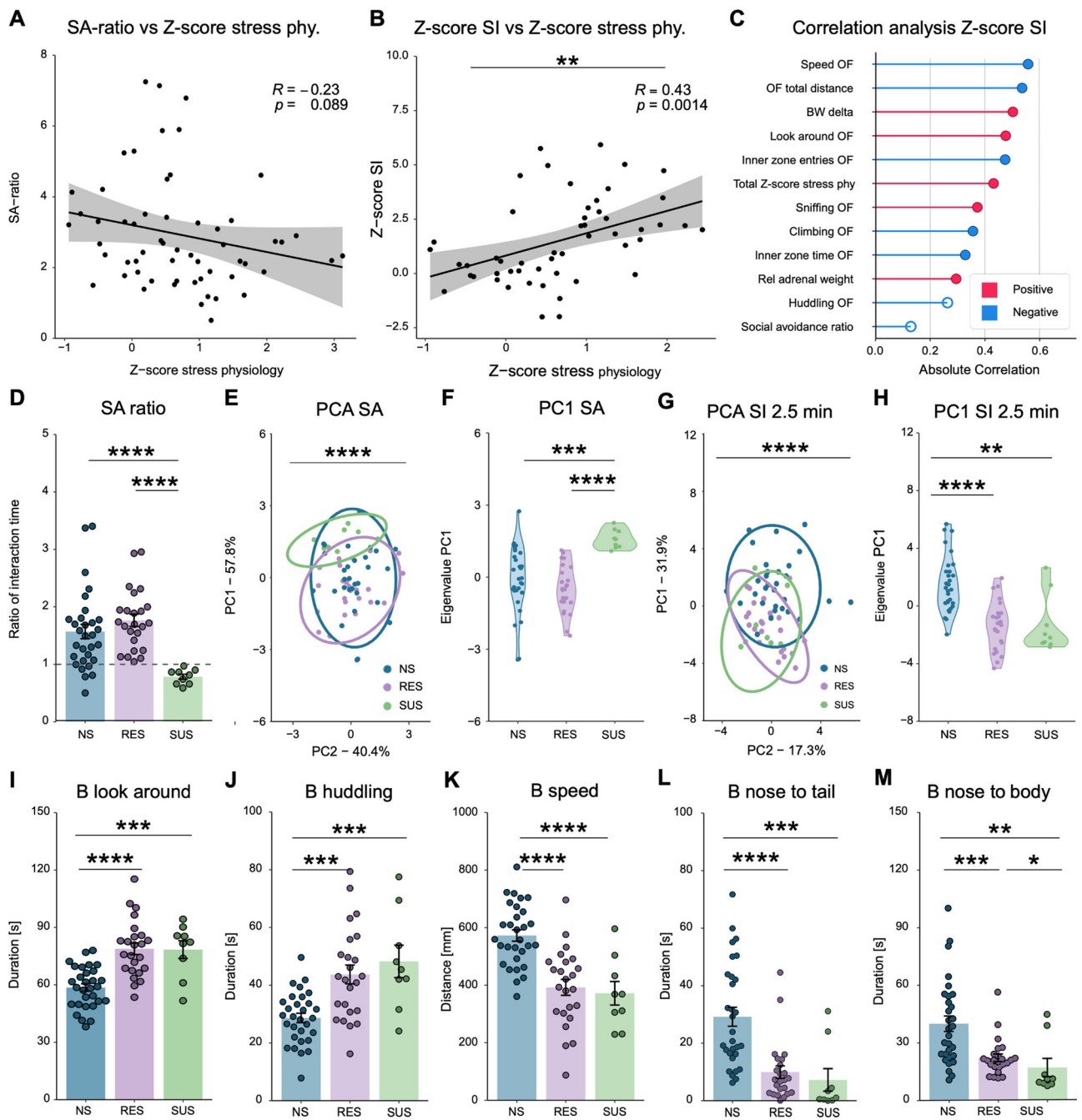

consecutive, non-overlapping bins retaining the optimal size was also reported (Fig. 6C and Supplementary Fig. 5C). Here, decaying performance across bins in the SI setting is also compatible with a state of arousal, where conditions become less distinguishable over time after the behavior of the C57Bl/6N mice becomes less influenced by novelty. The largest difference between NS and CSDS animals can thus be observed during this period. In line with this finding, the optimal distance in the single animal OF data was reached at 595 s, suggesting that no binning is necessary since behavior between conditions remains consistently distinguishable across the videos (Supplementary Fig. 6C).

Interestingly, global animal embeddings show a clearer separation between conditions in both single and multi-animal embeddings for the SI setting (Fig. 6D and Supplementary Fig. 5D), whereas the difference is milder in the OF setting, as the projected distributions are less separable (Supplementary Fig. 6D). In the SA setting, projections

show, as expected, a higher separation between conditions in trial two, which includes the encaged conspecific (Supplementary Fig. 7C, D).

These global embeddings also capture how distributions merge over time in the SI settings, as the behavioral profiles of NS and CSDS mice become closer (Supplementary Figs. 8, 9).

## Individual unsupervised clusters reveal differences in behavior enrichment

Going beyond global differences in behavior, the aggregated embeddings depicted so far are the result of summarizing the expression of the set of detected behavioral clusters. Once obtained, DeepOF enables the user to test the differential expression between conditions. To this end, the time spent on each cluster across all videos for each condition is recorded for each time bin. Importantly, DeepOF has no knowledge of the assigned animal conditions at the time of training and assigning clusters.

**Fig. 5 | Z-score correlation analysis and the exploration of susceptibility and resiliency. A** Pearson correlation analysis between the SA-ratio and the Z-score of stress physiology ($R = -0.23$, $p = 0.089$). **B** Pearson correlation analysis between the SI task 2.5 min time bin top five contributing behaviors and the Z-score of stress physiology ($R = 0.43$, $p = 0.0014$). **C** Pearson correlation analyses between the Z-score of SI and all other parameters. A strong correlation was observed with several OF parameters, such as speed ($R = -0.56$, $p = 1.76e-5$), total distance ($R = -0.54$, $p = 4.27e-5$), look-around ($R = 0.48$, $p = 0.0004$), and inner zone: entries ($R = -0.47$, $p = 0.0004$), but not with the SA-ratio ($R = -0.13$, $p = 0.37$). **D** The SA-ratio shows a significant main effect with the Kruskal-Wallis: $H(2) = 21.22$, $p < 0.0001$). Wilcoxon posthoc shows that SUS animals (SI-ratio <1) have a significantly lower SI-ratio compared to NS animals $W(9,30) = 249$, $p = 4.1e-5$ and RES animals $W(9,24) = 216$, $p = 1.56e-7$. There is no difference between NS and RES animals $W(30,24) = 270$, $p = 0.12$. **E** The PCA for SA shows a significant main effect with the one-way ANOVA: $F(2,60) = 10.90$, $p = 9.19e-5$. **F** The PC1 eigenvalues of the SA show a significant difference between SUS and NS animals Post-hoc Benjamini Hochberg (BH): $T(9,30) =p = 0.0005$ and between SUS and RES animals $T(9,24) =p = 5.88e-5$. There is no significant difference between NS and RES animals $T(30,24) =p = 0.196$. **G** The PCA for the 2.5 min SI ratio shows a significant main effect with the Kruskal-Wallis: $H(2) = 24.83$, $p = 4.06e-6$. **H** The PC1 eigenvalues of the 2.5 min bin SI show a significant difference between NS and RES animals Post-hoc Wilcoxon: $W(30,24) = 92$, $p = 1.82e-6$, and between NS and SUS animals $W(30,9) = 41$, $p = 0.0015$. There is no difference between RES and SUS animals ($W(24,9) = 117$, $p = 0.736$). **I** B-look-around shows a significant main effect with the one-way-ANOVA: $F(2,60) = 19.23$, $p = 3.53e-7$. Post hoc BH shows a significant difference between NS and RES ($T(30,24) =p = 9.86e-7$), and NS and SUS ($T(30,9) =p = 0.0002$), but no difference between RES and SUS $T(24,9) =p = 0.94$.

**J** B-huddle shows a significant main effect with the one-way-ANOVA: $F(2,60) = 12.35$, $p = 3.23e-5$. Post hoc BH shows a significant difference between NS and RES ($T(30,24) =p = 0.0003$), and NS and SUS ($T(30,9) =p = 0.0004$), but no difference between RES and SUS ($T(24,9) =p = 0.39$. **K** B-speed shows a significant main effect with the one-way-ANOVA: $F(2,60) = 18.63$, $p = 5.1e-7$. Post hoc BH shows a significant difference between NS and RES ($T(30,24) =p = 3.12e-6$), and NS and SUS ($T(30,9) =p = 7.62e-5$), but no difference between RES and SUS $T(24,9) =p = 0.67$. **L** B-nose-to-tail shows a significant main effect with the Kruskal-Wallis: $H(2) = 26.70$, $p = 1.59e-6$. Post hoc Wilcoxon shows a significant difference between NS and RES ($W(30,24) = 628$, $p = 1.82e-6$), and NS and SUS ($W(30,9) = 236$, $p = 0.0005$), but no difference between RES and SUS $W(24,9) = 152.5$, $p = 0.075$. **M** B-nose-to-body shows a significant main effect with the Kruskal-Wallis: $H(2) = 19.61$, $p = 5.52e-5$. Post hoc Wilcoxon analysis shows a significant difference between NS and RES ($W(30,24) = 567$, $p = 0.0003$), and NS and SUS ($W(30,9) = 230$, $p = 0.0009$), and RES and SUS $W(24,9) = 167$, $p = 0.018$. The correlation analyses (A, B) are represented with a regression line and a 95% confidence interval window and all individual samples as points. **C** has the correlation value ($R$) represented by the red line (positive) or blue line (negative), black circles around the points are identified as significant correlations, $p < 0.05$. The bar graphs are presented as mean ± standard error of the mean and all individual samples as points. The PCA graphs (E, G) are provided with a 95% confidence ellipse and all individual samples as points. Further PC1 analyses are represented with a violin plot and all individual samples as points (F, H). The bar graphs are presented as mean ± standard error of the mean and all individual samples as points. $N = 30$ for NS and CSDS in (A), and $n = 26$ for NS and $n = 27$ for CSDS in (B, C), $n = 30$ for NS, $n = 24$ for RES, $n = 9$ for SUS in (D–M). Source data are provided as a Source Data file.

The expression between NS and CSDS animals was then compared using 2-way Mann-Whitney $U$ tests for each cluster independently, and $p$ values were corrected for multiple testing using the BH method across both clusters and time bins, when applicable. We observed significant differences in eight out of ten and six out of ten clusters for the first time bin of the single and multi-animal SI settings, respectively (Fig. 6E and Supplementary Fig. 5E). Interestingly, and in line with habituation to the environment, these differences also fade across time. The single-animal setting still shows some (although less) significant differences in all time bins, albeit with reduced effect sizes (Supplementary Fig. 10). Interestingly, also in the single-animal embeddings, cluster 8 remains highly significant during the entire course of the experiments. The multi-animal setting yields in contrast almost no significant results beyond the first time bin (Supplementary Fig. 11).

In the OF setting, 7 out of 11 clusters showed a significant differential expression in the first 595 s (Supplementary Fig. 6E). The SA test, in turn, is an interesting setting to test DeepOF given that its main axis of variation is the distance to the cage with the conspecific, which constitutes information that is not available to DeepOF in its current form (which only looks at the posture of the tracked animals). Interestingly, and while the analysis shows no significant results in trial one (without the conspecific, Supplementary Fig. 7E), 6 out of 17 clusters show significant differential expression in trial two (with the conspecific, Supplementary Fig. 7F), suggesting that DeepOF can correctly detect behavioral differences even without absolute location information.

Finally, we also explored the spatial distribution of cluster expression across all three settings. We obtained heatmaps depicting the global exploration of the arena by the C57Bl/6N across all videos (for both conditions). Along these lines, our results show how, while, as shown, CSDS animals tend to occupy the center of the arena significantly less (Fig. 2F) there is no spatial preference across animals for individual clusters (Fig. 6F and Supplementary Figs. 5F, 6F show the overall locomotion distribution, while a comprehensive overview of individual clusters is presented in Supplementary Figs. 12, 13, and 14).

## Individual unsupervised clusters reveal differences in behavior dynamics

Aside from comparing cluster enrichment, DeepOF can help gain insight into how cluster transitions and sequences differ across conditions. To accomplish this, an empirical transition matrix was obtained for each condition by counting how many times an animal goes from one given cluster to another (including itself). Since all transitions were observed to have non-zero probability, the Markov chains obtained from simulations can be proven to reach a steady state over time (where probabilities to go from one behavior to another stabilize). The entropy of these steady state distributions was reported for both conditions, with higher values corresponding to a less predictable exploration of the behavioral space. Interestingly, CSDS animals showed a significantly lower behavioral entropy in the social interaction task than their NS counterparts, retrievable in both single and multi-animal embeddings (Fig. 6F and Supplementary Fig. 5F). This goes in line with the NS animals exploring the behavioral space more thoroughly, while CSDS animals are more conditioned by the conspecific. In line with this hypothesis, no significant differences across conditions were found in the single-animal OF experiments (Supplementary Fig. 6F). Moreover, to validate these results, the obtained behavioral entropy score was correlated with the physiology Z-score presented earlier (Supplementary Fig. 15). As expected, significant negative correlations were found for the SI setting both when exploring the single and multi-animal behavioral spaces. No significant correlation was observed for the single-animal OF setting.

## Shapley additive explanations reveal a consistent profile across differentially expressed clusters

An important aspect of any machine learning pipeline using highly complex models is its explainability. In this study, we aimed to explain cluster assignments by fitting a multi-output supervised classifier (a gradient boosting machine) that maps statistics of the initial time series segments (including locomotion and individual body part areas, speeds, distances, and angles) to the subsequent cluster assignments. Performance and generalizability of the constructed classifiers across the dataset were assessed in terms of the balanced accuracy on a 10-

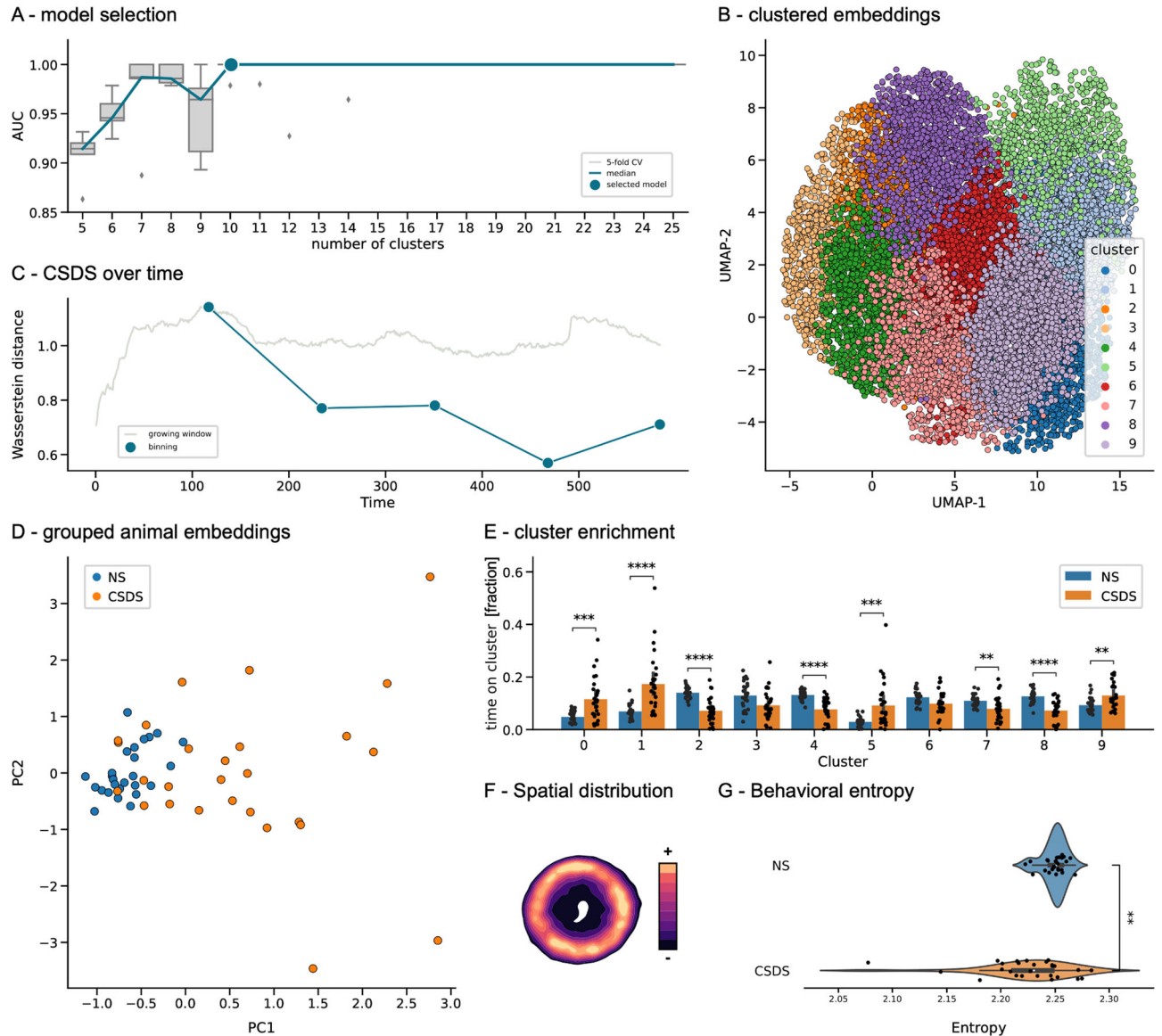

**Fig. 6 | Single-animal unsupervised analyses identify different behavioral patterns between stressed and non-stressed mice during the SI task. A** Cluster selection pipeline results reporting the area under the ROC curve from a logistic regression classifier discriminating between conditions. A 10-component solution (from a range between 5 and 25) was selected as optimal in a fivefold ($N = 5$) cross-validation loop (see methods for details). **B** Embeddings by time point obtained using DeepOF's unsupervised pipeline. Different colors correspond to different clusters. Dimensionality was further reduced from the original 8-dimensional embeddings using UMAP for visualization purposes. **C** Optimal binning of the videos was obtained as the Wasserstein distance between the global animal embeddings of both conditions across a growing window, between the first 10–600 s for each video at one-second intervals (gray curve). Higher values correspond to larger behavioral differences across conditions. A maximum was observed at 126 s, close to the stipulated 150 s selected based on the SA task literature. The dark green curve depicts the Wasserstein distance across all subsequent non-overlapping bins with optimal length. The decay observed across time is consistent with the hypothesized arousal period in the CSDS cohort. **D** Representation of the global animal embeddings for the optimally discriminant bin (126 s) per experimental video colored by condition (see methods for details).

**E** Cluster enrichment per experimental condition ($N = 26$ for NS and $N = 27$ for CSDS) in the first optimal bin (first 126 s). Reported statistics correspond to a 2-way Mann-Whitney U non-parametric test corrected for multiple testing using Benjamini-Hochbergs's method across both clusters and bins (significant differences observed in clusters 0: $U = 1.6e+2$, $p = 7.7e-4$, 1: $U = 1.1e+2$, $p = 1.3e-5$, 2: $U = 6.3e+2$, $p = 1.1e-6$, 4: $U = 6.4e+2$, $p = 3.3e-7$, 5: $U = 1.6e+2$, $p = 6.3e-4$, 7: $U = 5.3e+2$, $p = 1.3e-3$, 8: $U = 6.2e+2$, $p = 1.9e-6$, 9: $U = 1.9e+2$, $p = 4.4e-3$). Bar graphs represent mean ± standard deviation of the time proportion spent on each cluster. **F** Example heatmap depicting spatial distribution across all experiments (in both conditions) for all clusters. Specific heatmaps for all individual clusters are available in Supplementary Fig. 12). **G** Behavioral entropy scores per condition. NS animals show a significantly higher entropy than CSDS animals, which can be attributed to a less predictable exploration of the behavioral space ($U = 5.3e+2$, $p = 1.68e-3$, $N = 26$ for NS and $N = 27$ for CSDS). Moreover, and in accordance with these results, behavioral entropy shows a significant negative correlation with the presented stress physiology Z-score (Supplementary Fig. 15A). Source data are provided as a Source Data file. Box plots in (**A**, **G**) show the median and the inter-quartile range. Whiskers show the full range, excluding outliers as a function of the inter-quartile range.

fold stratified cross-validation loop, which was designed so that segments coming from the same video were never assigned to both train and test folds. Data for SI (single and multi-animal) and OF settings were standardized, and the minority class was oversampled using the SMOTE algorithm to correct for class imbalance. Performance per cluster is shown by means of the confusion matrices per task and the balanced accuracy per cluster (Fig. 7A, B and Supplementary Figs. 16A, B and 17A, B for all three settings, respectively). Importantly, classifier

### A - cluster detection confusion matrix

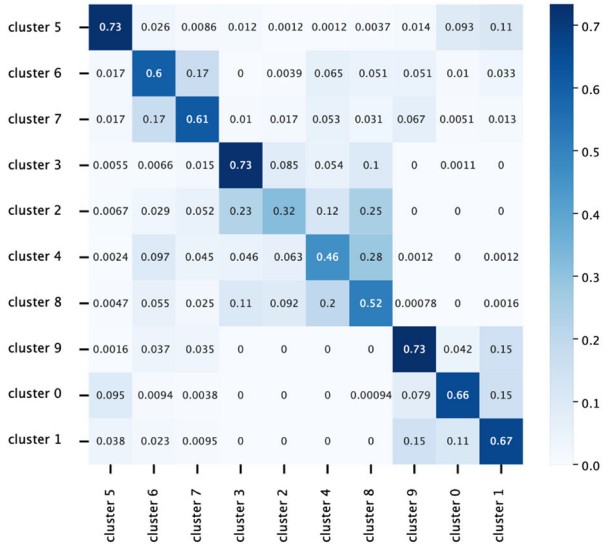

### B - cluster detection performance

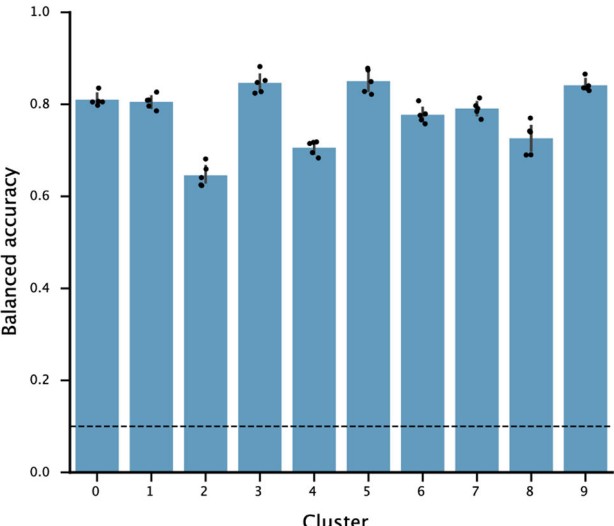

### C - SHAP global feature importance

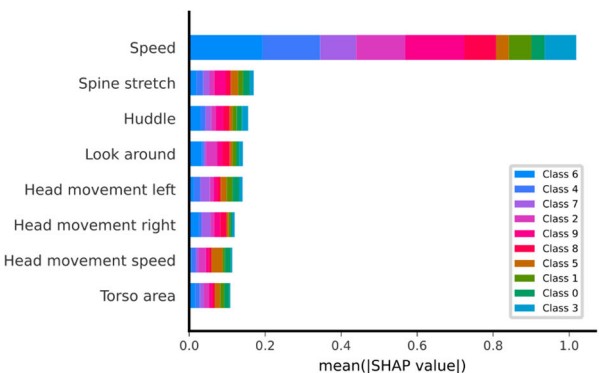

### D - SHAP analysis of single-animal SI cluster 1

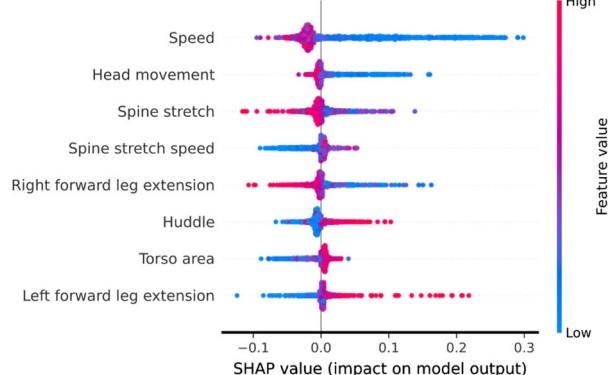

### E - SHAP analysis of single animal SI cluster 2

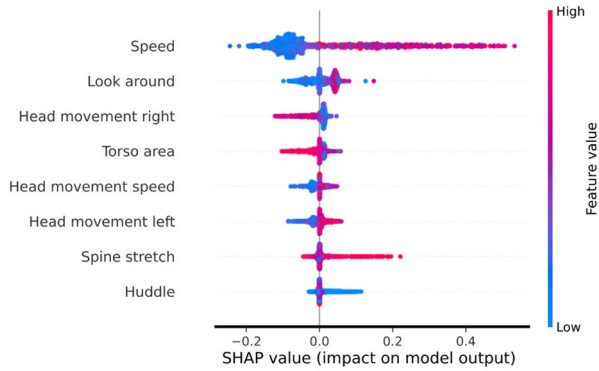

### F - SHAP analysis of single animal SI cluster 8

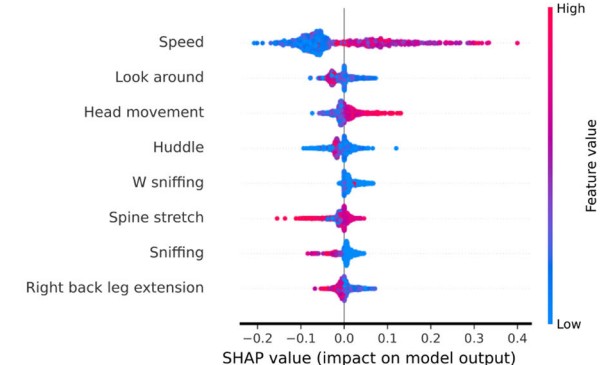

**Fig. 7 | SHAP analysis of unsupervised cluster assignments in the single-animal social interaction task.** Gradient boosting machines were trained to map from a predefined set of time series statistics (including body part speeds, distances, distance speeds, areas, area speeds, and supervised annotations) to the previously obtained cluster assignments. **A** Confusion matrix obtained from the trained gradient boosting machine classifying between clusters. Aggregated performance over the validation folds of a fivefold cross-validation is shown. **B** Validation performance per cluster across a fivefold ($N = 5$) cross-validation loop. Balanced accuracy was used to correct for cluster assignment imbalance. The dashed line marks the expected performance due to chance, considering all outputs. Bars show mean ± 95% confidence interval. **C** Overall feature importance for the multi-output classifier using SHAP. Features in the y-axis are sorted by overall absolute SHAP values across clusters. Classes on the bars are sorted by overall absolute SHAP values across features. **D**–**F** Bee swarm plots for the three most differentially expressed clusters between NS and CSDS mice (1, 2, and 5), identified with the unsupervised DeepOF pipeline on the SI experiments using single-animal embeddings. The depicted plots display the first eight most important features for each classifier, in terms of the mean absolute value of the SHAP values. Source data are provided as a Source Data file.

performance is substantially greater than random in all cases for all three settings, meaning that all clusters are highly distinguishable from one another by the set of summary statistics employed.

The result of this analysis is thus a set of feature explainers for each retrieved cluster, which can be used to interpret, alongside visual inspection of the corresponding video fragments (included as Supplementary files), what the obtained behavioral motifs represent. Both global (Fig. 7C, Supplementary Figs. 16C, 17C) and cluster-specific feature importance values can be retrieved. In this context, we found consistent descriptions of clusters that are differentially represented across conditions for all three tasks.

In the single-animal SI task, for example, cluster 1 (Fig. 7D, enriched in CSDS animals) is consistently explained by low locomotion speed, low head movement, and low spine stretch, and is positively associated with the huddle classifier. Visual inspection reveals a behavior close to freezing. Cluster 2 (Fig. 7E, enriched in NS animals) is in contrast explained by high locomotion speed, exploratory behavior, low head movement, and spine stretch. Close visual inspection depicts active locomotion and engagement with the conspecific. Interestingly, cluster 8 (Fig. 7F, enriched in NS animals across all time bins) is explained by increased speed, head movement, and negatively associated with sniffing. Visual inspection suggests engaging in motion (shifting from a still position to active locomotion).

In the case of the multi-animal SI setting, the explainability pipeline reveals how the models work differently when taking both animals into account. In this case, the two-animal system is embedded as a whole, and features including both animals are considered when running SHAP. As mentioned in the methods section, a regularization hyperparameter allows the system to focus more on interactions between the animals or in joint individual behaviors. In this case, we used a moderated value of the parameter that enables the contribution of both, which becomes apparent when analyzing the explainability profiles of the retrieved behaviors. Cluster 3, for example (Supplementary Fig. 16D, highly enriched in CSDS), is explained not only by low speed on the C57Bl/6 N animal, but also by increased speed of the CD1, among others. Upon visual inspection, one can observe exactly that the CD1 is exploring the arena while the C57Bl/6N stands still, in a posture usually associated with the stopped and huddled trait. Cluster 5 (Supplementary Fig. 16E, also enriched in CSDS) closely captures an interaction between the two animals, where the CD1 is typically more engaged in movement. The SHAP pipeline eloquently reveals negative correlations with spine stretch and back, torso, body and head areas, as well as speed in both mice. Conversely, cluster 8 (Supplementary Fig. 16F, enriched in NS) is well explained by increased speed in both animals, which can be confirmed by visual inspection.

Finally, this pipeline was also used to interpret clusters in the OF setting. In this case, cluster 0 (Supplementary Fig. 17D, enriched in CSDS animals) is explained by a decreased overall speed, positive correlations with mid and back spine stretch, back area, and left leg extensions, and negative association with right leg extensions. Visual inspection indeed reveals a cluster highly enriched in digging. Cluster 8 (Supplementary Fig. 17E, also enriched in CSDS animals), is in turn explained by decreased speed, mid, and back spine stretch, increased head area and extended right legs. Visual inspection shows a cluster enriched in slow walking, often including head movement and interaction with the walls. Finally, cluster 9 (Supplementary Fig. 17F, enriched in NS animals) is positively correlated with speed and head movement, and negatively correlated with spine stretch, among others. Visual inspection depicts an exploratory behavior with active movement.

All in all, the provided cluster explainability pipeline is a useful tool to interpret all reported patterns. Moreover, visual inspection of cluster snippets is also made possible with a single command within DeepOF, which makes the interpretation process more effective.

## Discussion

For decades there has been a trend to standardize and simplify social behavioral tests, which has led to an oversimplification of the description of the social behavioral repertoire. The current developments of open-source markerless pose estimation tools for tracking multiple animals have provided the possibility for more complex and socially relevant behavioral tests. The current study provides an open-source tool, DeepOF, which can investigate both the individual and social behavioral profiles in mice using DeepLabCut-annotated pose estimation data. Applying this tool, the current study identified a distinct social behavioral profile following CSDS using a selection of five traits annotated by DeepOF on the C57Bl/6N animal. In addition, a similar social behavioral profile was identified using an unsupervised workflow, which could detect behavioral differences in different experimental settings, including social interaction and single-animal open field tests, and a social avoidance task. Moreover, DeepOF allowed to study behavioral dynamics in unprecedented detail and identified the 5 min during the interaction with a novel conspecific as crucial for the social profiling of CSDS exposure in both supervised and unsupervised workflows. Overall, this study demonstrates the high utility and versatility of DeepOF for the analysis of complex individual and social behavior in rodents.

### DeepOF as part of a markerless pose estimation toolset

The initial release of DeepLabCut in 2018[29] provided a reliable and accessible tool for researchers around the globe to process markerless pose estimation data, which has undoubtedly changed the field of behavioral neuroscience. This has set in motion a rapid growth of tools for analyzing pose estimation data that are increasing the range of possibilities in the field, which were unimaginable using classical tracking approaches or manual scoring. An important distinction between these pose estimation analysis tools is whether they intend to extract pre-defined and characterized traits (supervised) or to explore the data and extract patterns without external information (unsupervised). The DeepOF module is designed to provide both analysis pipelines. The supervised behavioral classifiers offer a quick and easy-to-use analysis to detect individual and social behavioral traits without manual labeling. In addition, when differences between the conditions are not reflected in these traits, or the researcher aims to obtain behavioral embeddings, the DeepOF package can encode the data in a time-aware way that can report differentially expressed patterns in an unsupervised manner, taking single and multi-animal inputs.

### The supervised framework: spotting recognizable patterns

The supervised pipeline within the DeepOF package can be used on single and dyadic behavioral data in multiple-shaped arenas. DeepOF is capable of reporting a pre-defined set of behavioral traits without any extra labeling or training required. To accomplish this, it relies on both simple rule-based annotations and machine learning binary classifiers whose generalizability has been tested, trading off flexibility for ease of use. This makes it user-friendly for researchers without computational expertise to apply this supervised pipeline, without having to make any modifications. To further detect unsupported patterns, using a more involved and flexible tool (such as SimBA[37] or MARS[27]) could be a reasonable next step to take. These tools include a supervised approach that requires the user to label and train classifiers, providing the freedom to train powerful classifiers and recognize behavioral traits, which is especially beneficial for labs without computational expertise. However, in contrast to DeepOF, this approach also delegates to the user the responsibility of testing the generalizability of the results (how well the trained models can be applied to newly generated data, even in similar settings), which requires careful practices from the experimenters.

## The DeepOF module provides a more complete social behavioral profile than the social avoidance task

The social behavioral profile in CSDS-subjected animals has been measured extensively using the SA task, which is based on the separation of social behavioral traits between non-stressed and stressed animals[11,17,38]. Previous research has shown that rodents have a social interaction preference towards a novel conspecific compared to a familiar conspecific[39]. However, the duration of this social behavioral arousal state has not been well documented. In this context, and by replicating the time the SA task typically lasts for[10], the current study shows that the CSDS-related social behavioral profile, obtained with the DeepOF supervised classifiers, was increasingly observed during the first 2.5 min of the 10 min SI task. Furthermore, the presented unsupervised workflow was used to determine an optimal binning of our experiments by measuring how different both conditions were across time for a linear classifier. This yielded an optimal separation at ~2.1 min (126 and 124 s when testing with single and multi-animal embeddings, respectively), which then decayed over subsequent time bins in a manner consistent with the arousal hypothesis. The fact that this result was not seen in the absence of a conspecific strengthens this argument. Taking this into account, we argue that the introduction of a novel conspecific induces a state of arousal, which coincides with a distinct social behavioral profile that disappears over time after 2–3 min due to habituation.

Along these lines, this study shows that the DeepOF social behavioral classifiers provide a stronger separation of the social behavioral profile between stressed and non-stressed animals compared to the classical SA task, which also correlates better to physiological stress parameters.

Furthermore, the identification of stress-susceptible and resilient animals is often performed using the SA-ratio of the SA task[10,17] and for this DeepOF offers unique advantages. While the SA ratio clearly distinguishes stress-affected individuals, especially following more severe CSDS paradigms, the DeepOF module will significantly advance the possibilities and sensitivity of this distinction, by investigating the degree of resilience based on multiple behavioral classifiers with high sensitivity and in freely moving animals, which enables uncovering a so-far undescribed set of resilience-linked phenotypes that are different from the univariate SA task. Taken together, it can be concluded that using the DeepOF social behavioral classifiers provides a more robust and clearer social behavioral profile in animals subjected to CSDS compared to the SA task. An important reason for the superiority of DeepOF in social behavioral profiling depends on the experimental set-up: the SA task relies on the confinement of an animal (for example using a wired mesh cage), which means that no natural interaction between freely moving animals is possible, whereas the SI task is based on a naturally occurring interaction between freely moving animals[18]. Moreover, in the SA task, the confined animal can show symptoms of anxiety-related behavior, which influences the physiological state and the social interaction and approach behavior of the conspecific[40–42]. Differences in anxiety-related behavior between experimental animals can still contribute to alterations in social behavior and recent data suggest distinct neurobiological circuits driving both phenotypes[43], therefore sufficient habituation and the ability to observe behavior in freely moving animals will lead to improved discrimination. Moreover, a further crucial advantage of the DeepOF module is the many different behavioral classifiers that can be investigated at the same time without increasing the labor intensity. The combined analysis of multiple behavioral classifiers into a $Z$-score of social behavior provides a more complete social behavioral profile than solely investigating social avoidance behavior.

## DeepOF can detect and explain differences across experimental conditions in a fully unsupervised way, embedding data from one or more animals

The supervised pipeline within DeepOF follows a highly opinionated philosophy, which focuses on ease of use and relies on predefined models. As an alternative, DeepOF offers an unsupervised workflow capable of representing animal behavior across experiments without any label information. In its most basic expression, this involves obtaining a representation for each experiment in a time-aware manner: unlike other dimensionality reduction algorithms like PCA, UMAP, and T-SNE[26], DeepOF, when applied to the raw dataset, relies on a combination of convolutional and recurrent neural networks capable of modeling the sequential nature of motion. Each input to the models consists of a subsequence across a non-overlapping sliding window of each experiment. Although this idea has been explored before[33], DeepOF introduces several novelties to the field, such as unified embedding and clustering, the support for multi-animal embeddings, and graph representations that integrate not only coordinates by also body-part-specific speed and distance information, which makes it ideal for settings where informative body parts (such as paws) are occluded, as is the case for commonly used top-down videos.

In addition, these global embeddings can be decomposed into a set of clusters representing behavioral motifs that the user can then inspect both visually and with machine learning explainability methods. Moreover, by comparing cluster enrichment and dynamics across conditions, it is possible to answer questions that are relevant to understanding what the observed difference might be based on, without any previous knowledge: Which behaviors are most or least expressed in each condition? Is the set of behaviors expressed differently in experimental conditions? Are they expressed differentially across space and time? This constitutes a complementary approach that can be beneficial to further direct hypotheses when little knowledge is available. In addition, by not only showing overall differences between cohorts but also reporting which motion primitives might be driving them, it is possible to test hypotheses by training novel supervised classifiers based on those motion primitives. This can allow researchers to distinguish new, meaningful patterns that have not been reported before and that may be significantly associated with a given condition.

Taken together, the current study exemplifies that the unsupervised pipeline provided in DeepOF does not only recapitulate results previously obtained with the supervised analysis, but also shows how this tool can be used to detect habituation and overall differences in behavioral exploration. We also show that detected differences are significantly stronger when a conspecific is present, although also detectable during single animal arena exploration alone.

## Towards an open-source behavioral analysis ecosystem

One of the main advantages of DeepOF, SimBA[37], VAME[33], MARS[27], and many other packages cited in this manuscript, is that they are open source. This means that their inner workings are transparent, and that it is possible for the community to contribute to their development. We strongly believe that the adoption of open-source frameworks can not only increase transparency in the field but also incentivize a feeling of community, in which researchers and developers can share ideas, code, and solve bugs and problems together. Moreover, the open source framework facilitates beneficial feedback loops, where the data generated using these tools can be published, thus increasing the opportunity to produce better software. A good example of this is zero-shot pose estimation[44], which enables motion tracking without labeling, by cleverly leveraging information from several publicly available datasets. In addition, new technologies are starting to enable joint learning from multiple modalities, such as neural activation and behavior[45], which enables the exploration of how these modalities are influencing each other.

In addition to the software, an equally important problem to tackle is the need for open-source benchmarks. As platforms for testing and validating pose estimation and detection algorithms become available, it becomes easier to clearly show and compare the performance of different software options for different tasks. An

example of this is the Caltech Mouse Social Interactions (CalMS21) dataset, a pioneer in the field that provides benchmarking for classic detection of social interactions, annotation style transfer, and detection of rare traits[46]. While unsupervised learning benchmarking remains highly unexplored to the best of our knowledge, it would be crucial to compare the DeepOF pipeline with other available methods in this context when the tools become available.

Finally, and in contrast to several other options that offer extended functionality but rely on proprietary algorithms and/or specialized hardware[23], these tools have the potential to make otherwise expensive software available to a larger audience.

In conclusion, the current study provides a novel approach for individual and social behavioral profiling in rodents by extracting predefined behavioral classifiers and unsupervised, time-aware embeddings using DeepOF. Furthermore, while the tool provides means of customization, it is uniquely optimized for the most common behavioral setup: top-down video recordings. Moreover, we show evidence for the validation of the provided behavioral annotators and offer an open-source package to increase transparency and contribute to the further standardization of the behavioral constructs. We also show that, while differences across conditions are detectable during single animal exploration, they are enhanced in the SI task involving a companion mouse. Furthermore, while the classical SA task does identify the social behavioral profile induced by CSDS, the DeepOF behavioral classifiers provide a more robust and clearer profile. DeepOF is thereby a highly versatile tool that can also be applied to other research questions, e.g., to study sex differences in social behavior or analyze home-cage behavior throughout the lifespan of animals using longitudinal recordings. In addition, the DeepOF module contributes to a more specific classification of the affected individual and social behaviors in stress-related disorders, which could contribute to the study of drug development for psychiatric disorders.

## Methods

### Time series extraction from raw videos
Time series were extracted from videos using DeepLabCut version 2.2b7 (single animal mode). 11 body parts per animal were tagged, including the nose, left and right ears, three points along the spine (including the center of the animal), all four extremities, and the tail base (Fig. 1A). The DeepLabCut model was trained to track up to two animals at once (one CD1 mouse and one C57Bl/6 N mouse) and can be found in the Supplementary material (see code and data availability statement). Using the multi-animal DeepLabCut[30], extending the tracking to animals from the same strain is also possible. Next, DeepLabCut annotated datasets were processed and analyzed using DeepOF v0.4.6[36].

### Time series data preprocessing
All videos and extracted time series undergo an automatic preprocessing pipeline that is included within the DeepOF package, consisting of smoothing and two sequential imputation levels, applied to all body parts of all tracked animals independently. For smoothing DeepOF applies a Savitzky-Golay filter[47] to each independent tracked variable by fitting an n/2-degree polynomial over an n-frame sliding window, where n is the frame rate of the corresponding videos.

To identify and correct any artifacts in the time series, a moving average model is then fitted to the time-based quality scores of each tracked variable (as reported by DeepLabCut's output likelihood). By detecting divergences (of at least three standard deviations) from the moving average model, DeepOF can detect sudden and consistent drops in tracking quality, often correlated with body-part occlusions. Body parts with low quality are thus removed from the data, and further imputed using sci-kit learn's iterative imputer with default parameters[48], which predicts missing values based on all available

features at a given time point using a Bayesian ridge regression method. A second imputation method is then conducted, aiming to remove spatial jumps in the tracked body parts. To do this, another moving average model is fitted, this time to the body part coordinates themselves, and any data point located at least three standard deviations from the model is replaced by the predicted values.

### Time series feature extraction
After preprocessing the time series independently, DeepOF extracts a set of features aiming to describe how entire animals move and interact. These include centered and aligned coordinates, distances between body parts, angles, and areas of specific regions of each available body (Fig. 1B), as well as their speeds, accelerations, and higher-order derivatives. The value for each feature is reported per time point.

**Coordinates.** Raw coordinates for each body part are centered (the cartesian origin is set to the center of each animal) and vertically aligned so that the y-axis matches with the line delimited by the *center* of each animal and *spine 1* (see Fig. 1A for reference). This is done so that both translational and rotational variances are not considered in further processing steps (in principle, and except for some annotations such as wall climbing and sniffing—see below—DeepOF extracts posture patterns that are invariant to where in the arena and in which rotational orientation they are expressed).

**Distances and angles.** Distances and angles over time between all body parts within and across all animals are computed by DeepOF by default, and available for retrieval.

**Areas.** The full area of the animal over time is computed by DeepOF by defining a polygon on all external body parts (*nose, ears, legs,* and *tail base*). The head area is delimited by the *nose, ears,* and *spine 1*. The Torso area is delimited by spine 1, both *forward legs*, and *spine 2*. The back area is delimited by the *center*, both back legs, and the *tail base*.

Finally, speeds, accelerations, jerks, and larger-order derivatives of each extracted feature are also computed using a sliding window approach. Importantly, the detailed 11-body-part labeling scheme suggested and provided by DeepOF plays a crucial role here. While parts of the pipeline can still work with fewer labels, the comprehensive set of features that DeepOF is able to extract with this set of labels enhances not only supervised annotations, but also data representations and model interpretability.

### Supervised behavioral tagging with DeepOF
The supervised pipeline within DeepOF aims to provide a set of annotators that work out of the box (without user labeling) for several behaviorally relevant traits. The workflow supports both dyadic interactions and individual traits, which are reported for each mouse individually (Fig. 1C). Furthermore, annotated traits fall into one of two categories:

1. *Traits annotated based on pre-defined rules.* Several motifs of interest are annotated using a set of rules that do not require a trained model. For example, contact between animals can be reported when the distance between the involved body parts is less than a certain threshold.
2. *Traits annotated following a supervised machine learning pipeline.* While rule-based annotation is enough for some traits, others are too complex or might be manifested in subtly different ways, and machine learning models are often a better option. In this case, a rigorous validation pipeline has been applied to measure the performance of the classifier not only in a separate test data set, but also across datasets comprehending different arenas and laboratories.

**Rule-based annotated traits.** Among the rule-based annotated dyadic traits, nose-to-nose and nose-to-tail depend on single distance thresholds between specific body parts of the animals involved. In the case of nose-to-body, a single threshold is used between the nose of one animal and any body part of the other (except nose and tail base). Side-by-side and side-reverse-side are computed using two equal thresholds, measuring the distance between both noses and two tails in the former, and both nose-to-tail distances in the latter.

Of the individual traits, "look around" requires the animal to stand still (speed to be below a defined threshold) and the head to be moving (nose and ear speeds to be above a defined threshold). Finally, sniffing and wall climbing rely on the interaction of each animal with the arena (which can be detected automatically in certain settings, or indicated manually by the user using a GUI—graphical user interface—when creating a DeepOF project). An animal is annotated as sniffing the walls when speed is below a defined threshold, the distance between the nose and the wall is below a defined threshold, and the head is moving. Consequently, wall climbing is detected when the nose of an animal goes more than a certain threshold beyond the delimited arena. All mentioned thresholds can be specified (in millimeters) by the user. All analyses presented in this article were conducted with default values, which can be seen in Supplementary Table 1. Moreover, all annotations require a reported tracking likelihood of at least 0.85 on all involved body parts.

**Annotation using pre-trained machine learning models.** In the case of stopped and huddled, we trained a gradient boosting machine (scikit-learn, v1.2.0, default parameters) to detect the trait per frame, using a set of 26 variables including distances between body parts, speeds, and areas. Data were preprocessed by standardizing each animal's trajectories independently (controlling for body size), and the training set as a whole. Furthermore, to deal with the imbalanced nature of the dataset (as only 8.48 % of the frames were positively labeled) we applied Synthetic Minority Over-sampling Technique (SMOTE)[49] to oversample the minority class (using imblearn v0.10.1[50]).

Performance was then evaluated using a tenfold stratified cross-validation (to keep approximately the same number of positive labels in each validation fold) on a single dataset for model development and tested externally using a leave-one-dataset-out approach. Four independent datasets were used, collected in four different settings and across two different labs (see dataset details in Supplementary Table 2). Three of them (SI, OF, and SA) were tagged with manual labeling only, whereas the fourth (EX, obtained externally) combined manual labels and automatic pseudo-labeling using SimBA (Supplementary Fig. 2). The final classifier deployed with the latest version of DeepOF was then trained on a set of more than half a million labeled frames (567.367), coming from all four mentioned independent datasets, and global feature importance was obtained using SHAP (Shapley additive explanations).

After applying the annotators, a Kleinberg burst detection algorithm[37,51] is applied to all predictions. This step smoothens the results by merging detections that are close in time (called bursts) and removing isolated predictions, which an infinite hidden Markov model deems as noise. Moreover, rather than having a fixed detection window, the filter will be less likely to ignore isolated or less frequent events if they are far enough from higher frequency bursts but will be more prone to removing isolated events closer to a region where annotations are more frequent. In addition, it is important to notice that the annotators work independently, so more than one label can be assigned to an animal at a given time point (Fig. 1D).

Overall, while the provided behavioral set may not cover all scenarios, this out-of-the-box pipeline can be used to detect differences in behavior across experimental conditions without the need for further programming. More complex behaviors, involving user definition and labeling can thus be extracted using other available tools if required[37].

## Graph representations

To analyze complex spatio-temporal data involving features such as coordinates, speed, and distances, the unsupervised pipeline within DeepOF can structure the variables as an annotated graph (Fig. 1E).

In this representation, each node is annotated with three values, corresponding to both coordinates of each body part, as well as their speeds. Edges are in turn annotated with distances between both connected body parts. The adjacency matrix describing connectivity is provided by DeepOF for top-down videos, but can also be defined by the user. Moreover, this representation can be extended to a multi-animal setting, where independent graph representations for each animal are connected through nose-to-nose, nose-to-tail, and tail-to-tail edges, allowing the models to incorporate relative distances between animals. It is worth mentioning that the provided representation works best when adjacent body parts are being tracked so that propagation through space is not too coarse. One of the main assumptions behind spatio-temporal graph embeddings is that connected body parts are sufficiently correlated in space, which may not be the case if too little tracking labels are included[52].

## Unsupervised deep embeddings with DeepOF

Unsupervised analysis of behavior was conducted using an integrated workflow within DeepOF, which enables both the deep embedding of animal trajectories and their clustering, to retrieve motion motifs that are consistent across time.

To this end, node and edge features (for either single or multiple animals) are processed using a sliding window across time, and standardized twice: once per animal, to remove size variability, and a second time on the entire training set.

The resulting data is then embedded using a deep clustering neural network architecture based on Variational Deep Embeddings[53,54], a deep clustering algorithm that can be adapted to sequential data. During training of the models, DeepOF minimizes the ELBO (evidence lower bound), represented in Eq. (1):

$$L_{\text{ELBO}}(x) = \mathbb{E}_{q(z,c|x)}[\log p(x|z)] - D_{\text{KL}}(q(z,c|x)||p(z,c)) \qquad (1)$$

The first term corresponds to the reconstruction loss, which encourages the latent space ($z$) to represent the data ($x$) well over a set of clusters ($c$). The second term is the Kullback-Leibler divergence ($D_{\text{KL}}$) between a mixture-of-Gaussians prior ($p(z,c)$) and the variational posterior for each cluster ($q(z,c|x)$), which regularizes the embeddings to follow a mixture-of-Gaussians distribution where each component is associated with a particular behavior. A schematic overview of the model can be found in Fig. 1F.

Importantly, this loss function enforces a clustering structure directly in the latent space, removing the need for post-hoc clustering of the embeddings required by other available tools[33]. This has several advantages, the main one being that the clustering structure back-propagates to the encoder during training, improving clustering performance[55].

The main contribution of the provided architecture lies however in the encoder-decoder layers, which are designed to handle spatio-temporal graph data (in which connectivity is static, but node and edge attributes change over time)[56]. To accomplish this, features corresponding to each body part are first processed independently by a temporal block, which consists of a one-dimensional convolutional neural network (CNN) and two gated recurrent unit (GRU) layers. Subsequently, the outputs of these layers are passed by a spatial block, that shares information across adjacent body parts. This is accomplished using CensNet convolutions, a graph convolution architecture capable of embedding node and edge attributes at the same time[57]. This allows DeepOF to take advantage of several data modalities related to motion with a single data structure as input.

Once the models are trained, cluster assignments are obtained as the argmax of the posterior distribution given the data, as described in Eq. (2):

$$q(c|\mathbf{x}) = p(c|z) \equiv \frac{p(z)p(z|c)}{\sum_{c'=1}^{K} p(c')p(z|c')} \qquad (2)$$

where $c' \in (1, K)$ is an iterator over all clusters in the model.

In practice, this unsupervised pipeline can retrieve consistent patterns of animal motion in a flexible, non-linear, and fully unsupervised way. Moreover, as body part speeds and distances can be naturally included, this workflow works even when critical body parts (such as the paws) are occluded, which makes it ideal for top-down videos.

In addition, DeepOF is capable of training multi-animal embeddings by using multi-animal graphs (see graph representations section above). When more than one animal is detected, DeepOF allows the user to control how much these embeddings should consider interactions between the animals over the multi-animal system. This is achieved with an L1 penalization over the node embeddings in the aforementioned CensNet layers: larger values will prime the models to prioritize animal interactions, whereas smaller values will increase the contribution of the individual behavior of each animal. All experiments included in this study used a moderated parameter (0.25) which allowed the model to consider both interactions and joint individual behaviors.

### Unsupervised model training and hyperparameters

All unsupervised models used default values (as specified in DeepOF version 0.4.6). On each dataset, 10% of the available videos were used as a validation set to evaluate performance during training. Data were processed using sliding windows with a length matching the video frame rate of each dataset and stride of 1, mapping to eight-dimensional latent spaces. The training was conducted using the Nadam optimizer[58] (with a learning rate of 0.001 and gradient-based clipping of 0.75) over 150 epochs with early stopping based on the total validation loss and patience of 15 epochs. Upon training end, weights of the models are restored to those obtained in the best performing epoch using the same metric. The number of populated clusters over time, confidence in selected clusters (as the argmax of the produced soft counts), regularizers, and individual components of the loss function (see unsupervised deep embeddings with DeepOF section above) are tracked over time by DeepOF.

### Global animal embeddings

Aside from embedding time points individually, global animal embeddings (where each data point corresponds to the trajectory of an entire animal rather than to a single time point) were obtained by constructing a $k$-dimensional vector with the time proportion each animal spent on each cluster, where $k$ is the number of clusters in the given model.

### Cluster number selection

For each dataset that was analyzed with the unsupervised pipeline, models ranging from 5 to 25 clusters were trained five times, resulting in a total of 120 models per explored setting. All model hyperparameters were set to DeepOF defaults (see section below and API documentation for additional details). Global animal embeddings were then used as input to a logistic regression classifier (scikit-learn, default parameters) aiming to discriminate CSDS from non-stressed animals. The model with the smallest number of clusters that reached a performance within one standard deviation of the global maximum across the whole range (in terms of the area under the ROC—receiver operating characteristic—curve) was selected for further processing.

### Time binning and habituation quantification

A key aspect of DeepOF is that it allows for quantification of behavioral differences between cohorts over time in an unsupervised way. In this context, this is done by measuring the Wasserstein distance over time between the multivariate distributions describing global animal embeddings for CSDS and non-stressed animals.

By measuring this distance across a growing window, we can quantify how important additional information is to discriminate between conditions. This way, a peak in the distance curve would mark the point in time in which behavioral differences are maximized. In this study, we used a range between 10 and 600 s for each experiment, computing the Wasserstein distance between conditions every second. The time point at which the maximum was reached was selected as the optimal size for consecutive (non-overlapping) time bins. By reporting the behavioral distance along these bins, DeepOF can report behavioral habituation (which would involve behavioral differences between conditions decreasing over time).

### Unsupervised cluster interpretation using Shapley additive explanations (SHAP)

When applying the unsupervised pipeline, and quantifying which features DeepOF deems relevant for the unsupervised models to determine the assignment of a given time segment to a given cluster, all obtained sequence-cluster mappings were analyzed using Shapley additive explanations[59,60].

To this end, a comprehensive set of 52 distinct features (111 for two-animal embeddings) was built to describe each sliding window in the training set, including mean values of distances, angles, speeds, and supervised annotators.

Gradient boosting machines (using Catboost v1.1.1[61], which offers models specifically optimized for non-binary classification) were then trained to predict cluster labels from this set of statistics after normalization across the dataset and oversampling the minority class with the SMOTE algorithm[49]. Performance is reported as the validation balanced accuracy across a 10-fold stratified cross-validation loop, and feature importance (global and for each cluster) is reported in terms of the average absolute SHAP values, obtained using a permutation explainer.

### Animals for chronic social defeat stress experiments

Eight-week-old experimental male C57Bl/6N mice were bred in-house. The CD1 male mice (bred in-house) were used in the social avoidance and social interaction task as social conspecifics (CD1 animals were 4–6 weeks old) and as aggressors in the CSDS paradigm (CD1 animals were at least 16 weeks old). The study was conducted with male animals as a proof of principle, and for comparability to widely available data on chronic social defeat. All animals were housed in individually-ventilated cages (IVC; 30 cm × 16 cm × 16 cm connected by a central airflow system: Tecniplast, IVC Green Line—GM500) at least 2 weeks before the start of the experiment to allow acclimatization to the behavioral testing facility. All animals were kept under standard housing conditions; 12 h/12 h light-dark cycle (lights on at 7 a.m.), temperature $23 \pm 1\,°C$, humidity 55%. Food (Altromin 1324, Altromin GmbH, Germany) and water were available *ad libitum*. All experimental procedures were approved by the committee for the Care and Use of Laboratory Animals of the government of Upper Bavaria, Germany. All experiments were in accordance with the European Communities Council Directive 2010/63/EU.

### Chronic social defeat stress

At 2 months of age, male mice were randomly divided into the CSDS condition ($n = 30$) or the non-stressed condition (NS) ($n = 30$) (Supplementary Table 2, experiment code 1). The CSDS paradigm consisted of exposing the experimental C57Bl/6 N mouse to an aggressive CD1 mouse for 21 consecutive days, as previously described[62]. An additional

cohort (NS: $n = 30$, CSDS: $n = 33$, subdivided into susceptible animals $n = 9$, and resilient animals $n = 24$) was used to test the DeepOF social interaction classifiers on the resiliency and susceptibility division of the social avoidance ratio (Supplementary Table 2, experiment code 2). The prolonged 3-week CSDS paradigm was specifically chosen to elicit a more profound passive defeat phenotype, as originally reported by Kudryavtseva et al. [13], and to allow multiple behavioral assessments under stress conditions. In short, the CD1 aggressor mice were trained and specifically selected on their aggression prior to the start of the experiment. The experimental mice were introduced daily to a novel CD1 resident's territory, who attacked and forced the experimental mouse into subordination. Defeat sessions lasted until the stress-exposed mouse received two bouts of attacks from the CD1 aggressor or at 5 min in the rare instances when two bouts were not achieved within this duration. Animal health was monitored throughout the experiment to ensure that any minor injuries healed prior to the subsequent defeat session. Between daily defeats, stressed mice were housed in the resident's home cage but physically separated from the resident by a see-through, perforated mesh barrier, allowing sensory exposure to the CD1 aggressor mouse while preventing further attacks. The defeat time of day was randomized between 11 a.m. and 6 p.m. to avoid habituation and anticipatory behaviors in defeated mice. NS mice were single-housed in the same room as the stressed mice. All animals were handled daily and weighed every 3–4 days. Behavioral testing was performed after 14 days of the defeat paradigm, where behavior was observed in the morning and the defeat continued in the afternoon. The animals were sacrificed a day after the CSDS ended under deep isoflurane anesthesia by decapitation, which was at 3 months of age. Then, the adrenals were obtained, and the relative adrenal weight was calculated by dividing the adrenal weight by the body weight before sacrifice.

## Behavioral testing
Behavioral tests were performed between 8 a.m. and 11 a.m. in the same room as the housing facility. On day 15 of the CSDS paradigm, the animals were tested on the social avoidance (SA) task, while on day 16, the animals were tested on the combined open field (OF) and social interaction (SI) task. The SA task was analyzed using the automated video-tracking software AnyMaze 6.33 (Stoelting, Dublin, Ireland), whereas the OF and SI tasks were analyzed using DeepLabCut 2.2b7 for pose estimation [29,30], after which DeepOF module version 0.4.6 was used for preprocessing, supervised, and unsupervised analyses of behavior.

## Social avoidance
The SA task was performed in a square OF arena ($50 \times 50$ cm) to observe the social behavioral profile after CSDS, as well-established in previous studies [13,62–64]. The SA task consisted of two phases: the non-social stimulus phase and the social stimulus phase. During the non-social stimulus phase, which was the first 2.5 min, the experimental mouse was allowed to freely explore the OF arena with a small empty wired mesh cage against the wall of the OF. Then, the empty wired mesh cage was replaced with a wired mesh cage including a trapped unfamiliar young CD1 mouse (4–6 weeks old). During the following 2.5 min, the social-stimulus phase, the experimental mouse could freely explore the arena again. The SA-ratio was calculated by calculating the amount of time spent with the social stimulus, which was then divided by the time spent with the non-social stimulus. The identification of CSDS susceptibility and resiliency was obtained using a SA-ratio score of lower than "1" for susceptible animals, and an SI-ratio score higher than "1" for resilient animals.

## Open field and social interaction task
The OF and SI tasks were performed in a round OF arena (diameter of 38 cm). The bottom of the arena was covered in sawdust material to

minimize the cross-over effects of stress and anxiety by the novel environment. First, the OF task was performed, during which the experimental animal was allowed to freely explore the arena for 10 min. Subsequently, for the SI task, an unfamiliar young CD1 (4–6 weeks old) was introduced inside the arena and both animals were allowed to freely explore the arena for 10 min. The DeepOF module can identify five behavioral traits during the single animal OF task, which include wall-climbing, stopped-and-huddled, look-around, sniffing, and speed (locomotion), whereas in the SI task, all behavioral traits can be identified (Fig. 1C). During the analysis, the 10 min OF and SI tasks were analyzed in the total duration of the behavioral classifiers, and in time bins of 2.5 min to match the time frame in the SA task.

## Z-score stress physiology and social interaction calculation
The $Z$-scores combine the outcome of multiple tests via mean normalization and provide an overall score for the related behavior of interest. $Z$-scores were calculated as described previously[65]. The $Z$-score indicates for every observation ($X$), the number of standard deviations ($\sigma$) above or below the mean of the control group ($\mu$). This means that for each individual observation Eq. (3) is calculated:

$$Z = \frac{X - \mu}{\sigma} \tag{3}$$

Then, the obtained values need to be corrected for the directionality, such that an increased score will reflect the increase of the related behavior of interest. This means that per test, the scores were either already correct or were adjusted in the correct directionality by multiplying with "−1". Finally, to calculate the final z-score, the different z-scores per test were combined and divided by the total number of tests, as in Eq. (4).

$$Z_{total} = \frac{\sum_1^i z_{test_i}}{Number\ of\ tests} \tag{4}$$

The Z-score analysis of stress physiology is based on the relative adrenal weight and the body weight at day 21 of the CSDS, which are both strongly influenced by CSDS exposure[12]. The directionality of both tests did not require additional adjustment. Then, the Z-score of SI was calculated based on five DeepOF behavioral classifiers from the C57Bl/6N mouse, which were B-look-around, B-speed, B-huddle, B-nose-to-tail, and B-nose-to-body. The directionality was adjusted for B-speed, B-nose-to-tail, and B-nose-to-body.

## Behavioral entropy calculation
Shannon's entropy of the behavioral cluster space was obtained directly using DeepOF, as a measure of how predictable the sequence of behaviors expressed by a given animal is. To accomplish this, DeepOF obtains transition matrices across clusters using the unsupervised cluster assignments per animal. Stationary distributions for each transition matrix are then obtained by simulation through the matrices until convergence, and Shannon's entropy is computed for each stationary distribution. Entropy scores obtained for NS and CSDS animals were then compared. Overall entropy scores were also compared to the stress physiology Z-score for validation purposes.

## External dataset for validation of the DeepOF huddle classifier
An additional experiment was performed using different conditions and behavioral set-up, to assess the transferability of the DeepOF huddle classifier (Supplemental Table 2, experiment code 3) to data produced by a different lab. 12 weeks old C57BL/6J mice ($n = 24$, purchased from the Jackson Laboratory (catalog number 000664), Bar Harbor, ME, USA) were paired in a home-cage environment ($19 \times 19$ cm) with 12 weeks old ovariectomized CFW female mice

(purchased from Charles River Laboratories (catalog number 024), Wilmington, MA, USA) and were allowed to freely explore each other for 1.5 min. The animals were housed under standard laboratory conditions with a 12 h light–dark cycle (lights on from 07:00 to 19:00), temperature $22 \pm 1\,°C$, humidity 50%, in clear Plexiglas cages ($19 \times 29 \times 13$ cm) with unrestricted access to food (Purina Laboratory Rodent Diet 5001) and water. Procedures were approved by the McLean Hospital Institutional Animal Care and Use Committee and complied with the National Institutes of Health guidelines.

## Statistics

Statistical analyses and graphs were made in RStudio (R 4.1.1[66]) and python (v 3.9.13). All data were used for further statistical analysis unless stated otherwise. During the DeepLabCut tracking, seven animals were excluded due to technical difficulties (four NS and three CSDS were excluded). Statistical assumptions were then checked, in which the data were tested for normality using the Shapiro-Wilk test and QQ-plots and for heteroscedasticity using Levene's test. Data that violated these assumptions were analyzed using non-parametric tests. The time-course data was analyzed using the two-way ANOVA (parametric) or Kruskal-Wallis test (non-parametric) with time (days) as a within-subject factor and condition (NS vs. CSDS) as a between-subject factor, further posthoc analysis was performed using the Benjamini-Hochberg (BH) test (parametric) or the Wilcoxon test (non-parametric). P-values were adjusted for multiple testing using the Benjamini-Hochberg (BH) method. Three-group comparisons were analyzed using the one-way ANOVA (parametric) or Kruskal-Wallis test (non-parametric), and further posthoc analysis was performed using the BH test (parametric) or the Wilcoxon test (non-parametric). Two-group comparisons were analyzed using independent samples t-tests (parametric), Welch's tests (data are normalized but heteroscedastic), or Wilcoxon tests (non-parametric). Correlation analyses were performed using the Pearson correlation coefficient; outliers deviating more than 5 standard deviations from a fitted linear model were excluded from the analysis. The timeline and bar graphs are presented as mean ± standard error of the mean. Data was considered significant at $p < 0.05$ (*), with $p < 0.01$ (**), $p < 0.001$ (***), $p < 0.0001$ (****).

## Reporting summary

Further information on research design is available in the Nature Portfolio Reporting Summary linked to this article.

## Data availability

The authors declare that data supporting the findings of this study are available within the Article and Supplementary Information. Source data are provided with this paper.

## Code availability

All data and the accompanying code to perform the analyses and creating the figures are available for download via the Max Planck DataShare services. The most recent version of DeepOF is hosted in a GitHub repository, and a Zenodo release of the version used in this manuscript (v0.4.6) is found under https://doi.org/10.5281/zenodo.8013401. The most recent stable version of DeepOF is available in PyPI. Full documentation and tutorials are available on read the docs.

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

## Acknowledgements

The authors thank the DeepLabCut development team for creating and maintaining the DeepLabCut software. In addition, the authors thank Margherita Springer for the language editing of the manuscript and Max Pöhlmann for the design of the mouse illustrations in Fig. 2a. This study is supported by the Kids2Health grant of the Federal Ministry of Education and Research (01GL1743C, M.V.S.), and the European Union's Horizon 2020 research and innovation program under the Marie Skłodowska-Curie grant agreement (813533; L.M.).

## Author contributions

JB and MVS conceived the study. L.M. wrote the DeepOF module, with primary technical assessment from F.A., B.P., and B.M.M. J.B. and M.R. performed the experiments. L.M.B., L.v.D., C.E., L.D., and S.M. assisted with the experiments. J.B. and L.M. analyzed the data and wrote the first version of the manuscript. B.P. worked on figure design. J.B. created the mouse illustrations in Fig. 1. S.N., J.H., E.L.N., and K.J.R. assisted with manual behavioral data tracking and analysis for data benchmarking purposes. All authors contributed to the revision of the manuscript.

## Funding

## Competing interests

The authors declare no competing interests.
