## [Peer Review File · Nature Communications]

Automatically annotated motion tracking identifies a distinct social behavioral profile following chronic social defeat stressREVIEWER COMMENTS

Reviewer #1 (Remarks to the Author):

In this manuscript, Bordes et al., introduce the open source tool “Deep OF” to track distinct social behaviors during freely moving social interactions between mice that have or haven’t gone through chronic social defeat. The paper is well-written and the analysis pipeline is easy to follow for those without a background in machine learning. However, the existence of multiple similar tracking tools, and some methodological issues with the behavioral testing dampen enthusiasm. Detailed comments below:

Major Issues

1. The existence of tools such as Mouse Action Recognition System, Simple Behavioral Analysis and SLEAP, which perform overlapping functions, reduce the potential impact this tool may have on the field. Although these tools are mentioned in the discussion, it remains unclear what this tool adds beyond those that are already available.
2. The decision to run the freely moving interaction in an open field arena following an anxiety test in the same arena is concerning. The fact that defeated mice showed a decrease in total distance and inner-zone time suggests an anxiety-like phenotype. This may thus interfere with the animal’s willingness to engage the CD1 mouse regardless of their stress history.
3. Despite the extended period of social defeat, it does not seem to be very effective in producing social avoidance, a common behavioral endpoint of social defeat. The average SI ratio is 2 for defeated mice suggesting that very few, if any, mice were avoidant. This may also explain why there weren’t any differences in particular social behaviors throughout the entirety of the session.
4. There is no distinction between susceptible and resilient mice in terms of their behavioral profile during the social interaction. Although this may be due to the lack of susceptible mice discussed in the above comment, this is a crucial distinction that needs to be made in order to fit with the literature in general.

Minor Issues

1. The duration of each individual defeat, or the wounding criteria to end the defeat is not mentioned in the methods.
2. All figures should show individual data points, so the reader has an idea about the distribution of the data.

Reviewer #2 (Remarks to the Author):

The manuscript by Bordes et al., describes the development and use of an automatically annotated motion tracking, namely deepOF, that identifies distinct behaviors in single and individual mice following

chronic social defeat stress (CSDS). Unlike many other available options, DeepOF performs both supervised and unsupervised pipelines for assessing social behavior patterns. Both the results and the open-source tool are highly relevant and advance our understanding of the effects of CSDS on complex social behavioral patterns.

There are a number of minor comments that the authors need to address:

As DeepOF was able to identify novel behaviors in the open field test, why did the authors not use it to assess behavior in the social avoidance test, but only in the social interaction test? Could additional patterns be detected in the social avoidance test with DeepOF that would increase its ability to assess CSDS-relevant behaviors?

The novel behavioral pattern identified in the social interaction test are versus a young CD-1 mouse. Have the authors assessed how DeepOF reports same strain interaction or versus a female? If not, could they speculate whether DeepOF would be relevant for such situations?

Please provide more details and rationale for the choice of the 14 body parts that were marked. Additionally, do the authors, or have the, assessed if fewer would work as well, or if additional points could provide further information?

Why did the authors not track the CD-1 mouse with DeepOF and only the B6? Perhaps the CD-1 mouse displays behavioral patterns that influence those of the B6?

Line 148 - "merging detections close in time". Can the authors provide information about the timeframe meant, or number of video frames for example?

Line #89 - please provide more specific examples of the novel traits that are meant

Reviewer #3 (Remarks to the Author):

The manuscript by Bordes and colleagues describes a method (DeepOF) to analyse the behavioural patterns displayed by a mouse in the open field test, both in isolation and in combination with a different-coloured (white) conspecific. DeepOF builds on pre-defined features extracted from pose estimation point-tracking data, and it allows both supervised and unsupervised analyses. The supervised module is trained to identify 6 behaviours (wall-climbing, sniffing etc.), the unsupervised approach uses a variational autoencoder (VAE) to identify clusters of behaviour. The authors use a social defeat paradigm conducted in a large cohort of mice (n=30 per group) to test their algorithm. The supervised DeepOF is able to identify group differences in line with standard behaviour assessments, the unsupervised approach identifies clusters that can dissociate between treatment groups. The experiment and analyses are properly designed and conducted, and it shows that DeepOF (and by extension supervised and unsupervised machine learning methods in general) are better suited to study behaviour (in case of the CSDS paradigm). Similar demonstrations of the power of supervised and

unsupervised analyses have been published before. The main biological finding is that the standard way to assess the impact of social defeat, i.e. using a single social avoidance session, is less informative than the first 2.5 mins of a “free” social interaction test in the open field. This finding is interesting to stress researchers using the social defeat paradigm.

Major comments:

- The methods provided with this manuscript are insufficient to fully assess all aspects of the work, and insufficient to reproduce the work. The codebase is lacking sufficient documentation. The authors should provide a clear tutorial / readme. The current readme states that more detailed documentation, tutorials and method explanation will follow. This should have been taken care of before submission to allow for a more efficient/in-depth assessment. Similarly, the methods section needs to be much more detailed. It is unclear how the classifier(s) was/were trained for the supervised part of DeepOF. The methods need to describe the exact algorithm used (I assume it's some type of binary classifier due to some information that I only found in the discussion) including from what library/package and version number it was used, the choice of training, development, and test sets, the hyperparameters used for training and if/how they were tuned, how many epochs / iterations were used for training at what learning rate, the metrics used for accuracy assessment and many more details. Furthermore, it is unclear how manual annotation was performed (there is some information about this in supplementary figure 1, however this needs to be part of the methods). Beyond this example, many more details are missing from the methods section.

- It appears that the unsupervised method relies on both truly label free data (speed, areas, angles, distances etc), but also on the output of the supervised classifiers (= “extracted motion features”) that were trained/encoded manually. Unless I misunderstood, this is problematic in the sense that it cannot be considered an unsupervised method anymore at this stage. It depends on output from classifiers that were created with manually annotated data in a very much supervised fashion.

- It is unclear how the unsupervised method adds anything of value, because the p-values of the supervised “extracted motion features” (Figure 4) are better than the cluster p-values from the same dataset (Figure 6E).

- More context for the clusters should be provided. Can a human visually recognize some logic behind the clusters when looking at example sequences? Intuitively, one would expect these to correlate to “classical” behaviours as determined with the supervised module, since the underlying data are a mixture of unsupervised and supervised data. However, no such comparison is provided. The authors should investigate how their classical behaviours relate/map to the unsupervised cluster. For example, are multiple behaviours mapped to the same cluster or multiple clusters mapped to the same behaviour? How diverse are sequences that are contained in the same cluster? Are some clusters more likely to happen in certain areas of the behaviour setup? Are some clusters more likely to happen when the two animals are close?

- The fact that only one animal is tracked in a social interaction test is disappointing, given the advances

in multi-animal tracking. It would be interesting to take the behavior/position of the social partner into consideration. This would also make the paper more cutting-edge.

- Presumably the biggest concern is transferability. It is unclear how well DeepOF would perform in a different OFT set-up, with a different mouse strain or in a different lab altogether. The authors state that the ease of use and generalization of the method are its main advantage. However, it needs to first be demonstrated that the provided classifiers are transferable, or if they are subject to strong dataset specific biases. Even small differences in animal age / strain / testing environment can lead to classifiers that struggle to produce reliable data, not to mention mice whose behaviour has been more dramatically altered by an experimental manipulation (e.g. drug treatments etc.). These questions should at the very least be tested on more datasets from the same lab, ideally on datasets from other labs (several can be accessed publicly).

- Related to the previous point, could the algorithm be applied to the social avoidance test? This test is conducted in a similar size arena (albeit a square rather than a circle). Since most labs use square open field tests, this would already test some of the issues surrounding transferability.

- I did not find any online repository of the data. Neither video recording, nor pose-estimation data are accessible. It is necessary to deposit the raw video recordings for other labs to establish the approach using the same data, before testing it on their own data.

Minor comments:

- Throughout the results section a clearer distinction is needed between the OFT test (where the animal is tested alone: OFT-alone), and the OFT test where a second mouse was introduced (social interaction test). It was often unclear, which results pertain to which "test". This starts with a clearer visualization in Figure 2 when which test was run and which analyses in the subsequent figures relate to which test.

- Lines 342-344: The rationale for analysing time bins (and how those bins were chosen) is unclear. If time bins were analysed, why wasn't this done for the OFT-alone and the Social-Interaction datasets?

- Why was the social defeat stress run for 21 days, but behaviour only analysed after day 15? Were no behaviour tests conducted on day 21? Figure 2A suggests differently. This would be an important opportunity to test how reproducible the data are or how variable behaviour might still be between D15 and D21. It's a bit strange to correlate behaviour with many physiological measures taken a week later.

- In figure 5C, is the strongest correlation found for sniffing should be discussed. What might this mean? Is this sniffing against the wall as depicted in Figure 1C? An exploratory behaviour?

- Line 429: entropy is an interesting calculation. It's not clear, however, how this helps to "explain" animal behaviour. Does entropy in the social interaction data correlate with certain behaviours? Or with the social avoidance score? Or with the physiological stress score?

REVIEWER COMMENTS

Reviewer #1

In this manuscript, Bordes et al., introduce the open source tool “Deep OF” to track distinct social behaviors during freely moving social interactions between mice that have or haven’t gone through chronic social defeat. The paper is well-written and the analysis pipeline is easy to follow for those without a background in machine learning. However, the existence of multiple similar tracking tools, and some methodological issues with the behavioral testing dampen enthusiasm.

Response: We thank the reviewer for providing their insightful comments and helpful suggestions. Please find our detailed responses below.

Major Issues

1. The existence of tools such as Mouse Action Recognition System, Simple Behavioral Analysis and SLEAP, which perform overlapping functions, reduce the potential impact this tool may have on the field. Although these tools are mentioned in the discussion, it remains unclear what this tool adds beyond those that are already available.

Response:

The reviewer raises a relevant point that was not fully clear in the previous manuscript. The DeepOF toolbox provides a unique combination of supervised and unsupervised behavioral analysis using the same set of DeepLabCut annotated data. This contrasts with other available toolboxes, which provide either supervised analysis options, such as MARS (Segalin et al., 2021) and SimBA (Nilsson et al., 2020), or just unsupervised analysis options, such as VAME (Luxem et al., 2022). The SLEAP toolbox is, to the best of our knowledge, meant for pose-estimation tracking and does not integrate supervised or unsupervised behavioral analysis pipelines (Pereira et al., 2022).

Importantly, DeepOF provides a set of out-of-the-box supervised behavioral annotators that are specifically designed and validated for the most common behavioral set-up: top-down video recordings of up to two freely interacting mice. In this regard, the package follows an opinionated philosophy that aims to trade flexibility (since users can only apply the provided annotators) for ease of use (since no user labeling is required). Moreover, while the number of scientists exploring DeepLabCut pose estimation is on the rise, many have problems with data interpretation after the tracking. DeepOF is a unique tool that allows users to gather information about a set of individual and social interactions, without additional programming or data training needed. We believe it is worth mentioning that validation of these annotators has been significantly extended for this revised version of the manuscript, as depicted in supplemental figures 1 and 2.

On the other hand, DeepOF offers a deep clustering pipeline, capable of detecting differences in behavior across cohorts in a fully unsupervised way, which is a great exploratory alternative to the models mentioned above that also does not require labelling. Furthermore, the current version of the manuscript includes fully redesigned models that take advantage of graph representation learning to include individual body part speed and distances in a spatially coherent manner, making DeepOF ideal for settings where highly informative body parts such as paws are often occluded (as is the case with top-down videos). Importantly, users can modify the adjacency matrix of these graphs and adapt it to

other settings (such as bottom-up or side videos, for example), which makes DeepOF an extremely flexible tool in this regard.

Importantly, the version of the software presented in this revised manuscript also introduces multi-animal embeddings (as suggested by reviewers 2 and 3). By taking advantage of the aforementioned graph structure, joint individual behaviors and social interactions on two or more mice are now fully supported. The results presented in the manuscript show how, while the captured effects of CSDS are comparable across single and multi-animal embeddings, retrieved behaviors are specific to each pipeline. We believe this tool has strong potential to uncover behavioral differences in complex social systems outside CSDS.

Finally, the package does not only offer tools to annotate time series extracted from behavioral videos, but also to post-process, compare expression, dynamics, and consistency of the extracted patterns over time, as well as to interpret both supervised and unsupervised findings. Most figures and analyses presented in this manuscript (including statistics and supplemental videos) were created using DeepOF directly, illustrating how the tool can work in an end-to-end fashion. In addition, and from a software developing perspective, the package uses continuous integration and automatic testing, which helps quickly discover bugs when new features are released.

Furthermore, the relevance of DeepOF for the scientific community can also be seen by the rising number of groups that are already using the tool. At the moment, we have established its use in several internal and external collaborations, using not only CSDS but also other behavioral paradigms. Moreover, the software has been downloaded 1560 times at the time of writing this document, excluding mirrors and automatic testing on the cloud¹).

References to all these arguments have been added to the current version of the manuscript in the methods, results, discussion, and conclusions sections.

2. The decision to run the freely moving interaction in an open field arena following an anxiety test in the same arena is concerning. The fact that defeated mice showed a decrease in total distance and inner-zone time suggests an anxiety-like phenotype. This may thus interfere with the animal's willingness to engage the CD1 mouse regardless of their stress history.

Response:

We agree with the reviewer that anxiety in a novel environment may be an important factor impacting the displayed social behavior of animals in the test. Precisely for this reason, we allowed the test animal to explore the test arena for 10 minutes and habituate to this novel environment before the social partner is introduced. This habituation time is 4 times longer than in the conventional social avoidance test, where the animals only explore the novel open field arena for 2.5 minutes before the social conspecific is introduced under a wire mesh. In addition, the aversiveness of the arena of the social interaction test with freely moving mice was reduced by low lighting conditions and sawdust bedding, creating a home cage-like environment. Therefore, while the initial minutes of habituation to the novel environment can still be utilized to assess differences in anxiety between animals, the behavior of the

¹ Downloads retrieved from the public PyPI download statistics dataset provided by the Linehaul project through Google BigQuery. This approach allowed us to only quantify direct downloads, excluding inflation through mirrors and automatic testing on the cloud. At the time of writing, DeepOF has been downloaded 460 times from the python package index (pypi), 1051 times directly from the browser (through their GitHub and GitLab repositories), and 49 times via the provided Docker image. For comparison, the number of downloads without these filters rises to 15317.

animals clearly habituates over the course of the 10 minutes exploration. To better illustrate this point, we have now added the binned analysis of the 10-minute open field habituation as **new supplemental figure 3**. The data indicate sufficient habituation to the novel environment and ensures that the subsequent social behavior is not primarily driven by differences in anxiety due to exposure to the test arena. We now also clarified this point in the results section (**page 18-19, lines 500-510**) and discussion section (**pages 35, line 774-778**). However, it is also clear that chronic stress exposure increases anxiety-related behavior and that this is likely impacting also on the social behavior of the animals, although a recent paper indicated distinct neurobiological circuits driving both phenotypes (Morel et al., 2022). We therefore also revised the discussion to better illustrate this important point.

3. Despite the extended period of social defeat, it does not seem to be very effective in producing social avoidance, a common behavioral endpoint of social defeat. The average SI ratio is 2 for defeated mice suggesting that very few, if any, mice were avoidant. This may also explain why there weren't any differences in particular social behaviors throughout the entirety of the session.

Response:

We thank the reviewer for this important comment and agree that this should be emphasized better in the manuscript. We are confident that our chronic social defeat paradigm induced a robust stress effect, as we observe significant physiological hallmarks of stress (e.g. body weight and adrenal weight). Further, while the absolute levels of SI are somewhat higher than what was reported by others, the relative effect of SA-ratio reduction compared to controls is highly comparable or even stronger in the current data set (our data on SA-ratio: nonstressed (NS) mean: 3.41 and stressed (CSDS) mean: 2.38 = delta 1.03; literature example (Golden et al., 2011): NS mean: 1.45 and CSDS mean: 0.90 = delta 0.55). An additional difference to previous reports is the protocol applied in our study, where the length of social defeat interaction was limited by the number of attacks from the resident CD1 male and therefore often lasted shorter than 5 minutes. This further emphasizes the high sensitivity of DeepOF-based social phenotyping, which will enable to detect also more subtle alterations of social behavior.

To address this point more clearly, we now extended the discussion on this point and emphasize that our data do not exclude that stronger CSDS paradigms may also result in more long-lasting social behavioral alterations throughout the entirety of the test session (**page 27, lines 761-767**).

4. There is no distinction between susceptible and resilient mice in terms of their behavioral profile during the social interaction. Although this may be due to the lack of susceptible mice discussed in the above comment, this is a crucial distinction that needs to be made in order to fit with the literature in general.

Response:

We agree with the reviewer that a distinction of stress susceptible and stress resilient individuals is crucial and indeed DeepOF will significantly advance the possibilities and sensitivity of this distinction. While in many cases a cut-off of SA-ratio < 1 was defined to differentiate stress susceptible individuals, this cut-off is quite arbitrary and varies across experimental conditions and cohorts. For example, while in our initial cohort the SI baseline in non-stressed individuals is substantially higher than 1 (namely an average of 3.41), in other laboratories and settings the non-stressed SA-ratio baseline is only moderately higher than 1 (e.g., (Golden et al., 2011; Lorsch et al., 2021): average 1.45) or even close to 1 (e.g. (Labonté et al., 2019; Torres-Berrío et al., 2020)).

Independent of these cohort variabilities, DeepOF allows differentiating the degree of resilience based on multiple behavioral classifiers with high sensitivity and in freely moving animals, which enables uncovering a novel set of resilience-linked phenotypes. To illustrate this point, we have run a new social defeat experiment and analyzed social behavior with the social avoidance test and in freely moving mice using DeepOF analysis. The data for this cohort are now included as **new supplemental figure 6**. In this cohort the SI ratio in the social avoidance test of controls was at an average of 1.57, and CSDS animals could be differentiated into susceptible and resilient mice based on an SA-ratio cutoff of <1. Intriguingly, the DeepOF measurements obtained in mice interacting with a freely moving conspecific showed a robust and clear stress effect exactly in line with our previous data, but did not differ between susceptible and resilient animals identified in the social avoidance task. These results clearly indicate that analyzing social behavior in freely moving mice using DeepOF uncovers not just a more detailed behavioral phenotype compared to the social avoidance task but enables to identify additional subsets of animals that display distinct social behavioral features (i.e., susceptible mice) or are in their expression of social behavior indistinguishable from non-stressed controls (i.e., resilient mice). The DeepOF tool, therefore, enables researchers to assess the individual and social behavior of mice in a much higher detail and sensitivity, which will ultimately allow a better understanding of biological mechanisms underlying this behavior, including resiliency and susceptibility to stress. In the revised manuscript, we have now extended the results (**page 20, lines 561-567**) and discussion (**page 27, lines 761-767**) on this point.

Minor

Issues

1. The duration of each individual defeat, or the wounding criteria to end the defeat is not mentioned in the methods.

Response:

We thank the reviewer to bring this to our attention. We have revised the materials and methods (**page 14, 378-385**) in the manuscript to include the description of the defeat protocol.

2. All figures should show individual data points, so the reader has an idea about the distribution of the data.

Response:

We thank the reviewer for this important comment, this has now been revised in all figures.

Reviewer #2

The manuscript by Bordes et al., describes the development and use of an automatically annotated motion tracking, namely deepOF, that identifies distinct behaviors in single and individual mice following chronic social defeat stress (CSDS). Unlike many other available options, DeepOF performs both supervised and unsupervised pipelines for assessing social behavior patterns. Both the results and the open-source tool are highly relevant and advance our understanding of the effects of CSDS on complex social behavioral patterns.

Response:

We thank the reviewer for their positive and supporting evaluation of our work. We address all raised comments in detail below.

There are a number of minor comments that the authors need to address: As DeepOF was able to identify novel behaviors in the open field test, why did the authors not use it to assess behavior in the social avoidance test, but only in the social interaction test? Could additional patterns be detected in the social avoidance test with DeepOF that would increase its ability to assess CSDS-relevant behaviors?

Response:

We thank the reviewer for this excellent suggestion, which we now incorporated into the revised manuscript. We therefore used the unsupervised pipeline within DeepOF to analyze potential behavioral differences in both trials (Trial 1; without social conspecific and Trial 2; with social conspecific) of the social avoidance task. Interestingly, no different behavioral patterns were observed for the social avoidance trial 1 data between the non-stressed (NS) and stressed (CSDS) animals, however clear differences emerged in the social avoidance trial 2 data (see new supplemental figure 9). The data indicate that in addition to this validated measure of social avoidance, DeepOF enables the detection of additional behavioral alterations in the experimental animals, some of which might be even unrelated to the social avoidance per se and rather reflect additional behavioral disturbances induced by chronic stress exposure. We have now incorporated these results in the manuscript (**page 22-23, lines 628-640**).

The novel behavioral pattern identified in the social interaction test are versus a young CD-1 mouse. Have the authors assessed how DeepOF reports same strain interaction or versus a female? If not, could they speculate whether DeepOF would be relevant for such situations?

Response:

We thank the reviewer for this important point. Indeed, the investigation of female social behavior and same-strain social behavior are highly relevant research questions to investigate with the DeepOF tool. We are currently following this up in different studies and have already collected data demonstrating important behavioral differences in female-female social interactions as well as same strain male social interactions following CSDS. As these projects are ongoing and don't directly contribute to the validation of the DeepOF toolbox, they are not incorporated in the current manuscript. To address the reviewer's point, we now extended the discussion (**page 29, lines 849-851**) and illustrate how DeepOF-based behavioral profiling will aid these research approaches in the future.

Please provide more details and rationale for the choice of the 14 body parts that were marked.

Additionally, do the authors, or have they, assessed if fewer would work as well, or if additional points could provide further information?

Response:

We agree with the reviewer that the initial selection of body parts was not sufficiently well justified. In the process of the revisions, we have in fact now carefully examined the necessity of each of the body parts and consequently reduced the number of labels to 11, removing all three labels along the tail.

Regarding the removal of the labels along the tail, we found that their tracking was not consistently accurate, and their position was extremely variable, making unsupervised embeddings disproportionately rely on them. Additionally, these labels were not used at all for the supervised annotations.

Regarding the final set of 11 labels, it is worth noting that, while not all are used for rule-based annotation directly, they indeed take part in the features used to train the huddling classifier. Moreover, we would like to highlight that the enhanced unsupervised pipeline introduced with this version of the manuscript benefits from graph representations, which require adjacent body parts to be tracked for the adjacency matrix not to be too coarse (which would defeat the purpose of the spatial propagation of information across the graph). Finally, all 11 labels also provide rich insights when interpreting clusters using Shapley additive explanations. This has now been revised in the manuscript in the materials and methods (**page 7, lines 165-169, and page 10, lines 244-247**).

Why did the authors not track the CD-1 mouse with DeepOF and only the B6? Perhaps the CD-1 mouse displays behavioral patterns that influence those of the B6?

Response:

The reviewer raises a very valid point and a shortcoming in the previous DeepOF version. While the original reasoning behind single-animal tracking is that we have a single experimental subject (as only the C57Bl/6N animal was subjected to social defeat). However, building upon the provided suggestions, the revised version of DeepOF presented here is now compatible with multi-animal unsupervised embeddings, capable of tracking and retrieving patterns that take both animals (and potentially more) into account. By applying this pipeline, we were able to detect similar CSDS effects than with the single-animal embeddings, but driven by completely different behaviors involving both joint individual behaviors (such as a cluster where one animal is exploring and the other huddling, for example) and social interactions (such as a cluster representing the CD1 mice actively engaging in interaction with the black 6 is enriched in CSDS animals). All in all, we believe this is a useful tool that can open the door to embed more complex social systems.

Line 148 - "merging detections close in time". Can the authors provide information about the timeframe meant, or number of video frames for example?

Response: We are certainly glad to clarify any concerns and have added all more details to the methods' section of the manuscript.

The Kleinberg filter does not necessarily rely on a fixed detection window, but rather on an infinite hidden Markov model process in which, after each time point, the state of the system probabilistically determines how much time will pass until the next event (Kleinberg, 2002). To compute this probability, the gaps between events are assumed to be drawn from an exponential distribution with mean proportional to a user defined parameter s^i (where s is set to the number e in DeepOF by default).

Higher values of this parameter make it more unlikely for the model to report isolated events. The transition to higher order states is then proportional to another parameter gamma (set to 0.01 by default in DeepOF), higher values of which result in longer sustained bursts being needed for the algorithm to call a state change.

Roughly speaking, this means that the filter will be less likely to ignore isolated or less frequent events if they are far enough from a higher frequency burst but will be more prone to removing isolated events closer to a region where annotations are more frequent. This has now been revised in the manuscript in the materials and methods (**page 9, lines 222-227**).

Line #89 - please provide more specific examples of the novel traits that are meant

Response:

We have gladly added a more thorough description of the detected clusters to the results' subsection titled **"Shapley additive explanations reveal a consistent profile across differentially expressed clusters, see page 23, starting, lines 642-696"** for both single and multi-animal embeddings in the social interaction setting, and for the single-animal open field setting. A plethora of consistent behaviors were found to be differentially expressed between NS and CSDS in all three settings. Revisiting two examples from the manuscript:

- 1) In the single-animal SI task, for example, cluster 1 (figure 7D, enriched in CSDS animals) is consistently explained by low locomotion speed, low head movement, and low spine stretch, and is positively correlated with the huddling classifier. Visual inspection reveals a behavior close to freezing.
- 2) In the case of the multi-animal SI setting [...] cluster 5 (supplemental figure 17E, also enriched in CSDS) closely captures an interaction between the two animals, where the CD1 is typically more engaged in movement. The SHAP pipeline eloquently reveals negative correlations with spine stretch and back, torso, body and head areas, as well as speed in both mice.

Reviewer #3

The manuscript by Bordes and colleagues describes a method (DeepOF) to analyse the behavioural patterns displayed by a mouse in the open field test, both in isolation and in combination with a different-coloured (white) conspecific. DeepOF builds on pre-defined features extracted from pose estimation point-tracking data, and it allows both supervised and unsupervised analyses. The supervised module is trained to identify 6 behaviours (wall-climbing, sniffing etc.), the unsupervised approach uses a variational autoencoder (VAE) to identify clusters of behaviour. The authors use a social defeat paradigm conducted in a large cohort of mice (n=30 per group) to test their algorithm. The supervised DeepOF is able to identify group differences in line with standard behaviour assessments, the unsupervised approach identifies clusters that can dissociate between treatment groups. The experiment and analyses are properly designed and conducted, and it shows that DeepOF (and by extension supervised and unsupervised machine learning methods in general) are better suited to study behaviour (in case of the CSDS paradigm). Similar demonstrations of the power of supervised and unsupervised analyses have been published before. The main biological finding is that the standard way to assess the impact of social defeat, i.e. using a single social avoidance session, is less informative than the first 2.5 mins of a “free” social interaction test in the open field. This finding is interesting to stress researchers using the social defeat paradigm.

Response: We thank the reviewer for their constructive and helpful comments. Please see below our detailed responses.

Major comments:

- The methods provided with this manuscript are insufficient to fully assess all aspects of the work, and insufficient to reproduce the work. The codebase is lacking sufficient documentation. The authors should provide a clear tutorial / readme. The current readme states that more detailed documentation, tutorials and method explanation will follow. This should have been taken care of before submission to allow for a more efficient/in-depth assessment. Similarly, the methods section needs to be much more detailed. It is unclear how the classifier(s) was/were trained for the supervised part of DeepOF. The methods need to describe the exact algorithm used (I assume it's some type of binary classifier due to some information that I only found in the discussion) including from what library/package and version number it was used, the choice of training, development, and test sets, the hyperparameters used for training and if/how they were tuned, how many epochs / iterations were used for training at what learning rate, the metrics used for accuracy assessment and many more details. Furthermore, it is unclear how manual annotation was performed (there is some information about this in supplementary figure 1, however this needs to be part of the methods). Beyond this example, many more details are missing from the methods section.

Response:

We thank the reviewer for their extensive and detailed comment and agree that the description of the methods provided in the original manuscript was often insufficient and lacking detail. We have now fully revised these sections.

In detail, we:

- Thoroughly explained how time series are preprocessed within DeepOF (see section titled "Time series data preprocessing" in the methods, starting at line 123 in page 6).
- Explained in more detail how feature extraction works within DeepOF (see section titled "Time series feature extraction" in the methods, starting at line 140 in page 6).

- Greatly extended the methods and supplemental material explaining how the rule-based annotation of traits works, including a table with detailed rules and default parameters.
- Added a whole two sections to methods ("Annotation using pre-trained machine learning models") and results ("The supervised pipeline provided by DeepOF yields generalizable annotations") explaining how the huddling classifier was trained, and how its performance was evaluated both internally and with a newly collected external dataset (see also our response to the issue of transferability below). A supplemental figure to support our claims was also added (see supplemental figure 2).
- Explained how manual annotation was performed using Colabeler and SimBA, and added specific information on every dataset that was used.
- Added a section describing how now DeepOF includes graph representations of the input for the unsupervised pipeline, and how these enable multi-animal unsupervised embeddings and better performance in top-down videos.
- Rewrote the unsupervised pipeline section, updating the models we used (the previously used VQVAE models are still available within the package, but a better performing architecture based on VaDE is selected by default). We also explicitly included equations depicting the main loss function being minimized and the cluster selection process, and added a section with training hyperparameters (see "Unsupervised deep embeddings with DeepOF", starting from line 249 in page 10, and "Unsupervised model training and hyperparameters", starting from line 298 in page 11).
- Global embeddings and cluster number selection have their own sections now, and explanations are much more detailed (see "Global animal embeddings" starting from line 310 in page 12, and "Cluster number selection", starting from line 316 in page 12).
- Time binning and habituation quantification has its own section now (see "Time binning and habituation quantification", starting from line 326 in page 12).

As a side note, package versions were not specifically added unless in some special cases, as all used software is listed as a dependency of DeepOF v0.2, unless specified otherwise. Moreover, the documentation website has now been updated with three tutorials (covering the main functions for data preprocessing, supervised, and unsupervised workflows respectively) and a complete API reference. Taken together, we believe all the specific comments that the reviewer raised are now addressed. In a spirit of transparency, however, we are happy to accept more suggestions in case anything remains unclear or should still be expanded.

- It appears that the unsupervised method relies on both truly label free data (speed, areas, angles, distances etc), but also on the output of the supervised classifiers (= "extracted motion features") that were trained/encoded manually. Unless I misunderstood, this is problematic in the sense that it cannot be considered an unsupervised method anymore at this stage. It depends on output from classifiers that were created with manually annotated data in a very much supervised fashion.

Response:

We appreciate the reviewer's concern and want to clarify that indeed no supervised information is provided to the unsupervised models at any time during the pipeline. We understand the confusion may have arisen from the usage of supervised labels (alongside other features) in the post-hoc explainability pipeline applied once the models were trained and the clusters observed. In short, a set of features (including speed, areas, distances, and supervised annotations) is used to summarize the sliding windows that were originally fed to the unsupervised deep models, and a multi output classifier is then used to map them to the already assigned cluster labels in an unsupervised way. This allows us

to have a more interpretable mapping between features and clusters, able to detect associations with, among others, the provided supervised annotators.

To further clarify any confusion, we have extended the description of the pipeline in the methods (see "Unsupervised cluster interpretation using Shapley additive explanations (SHAP)", starting from line 340 in page 13) and results (see "Shapley additive explanations reveal a consistent profile across differentially expressed clusters", starting from line 642 in page 23). We hope this prevents further confusion and clarifies any remaining questions on the topic.

- It is unclear how the unsupervised method adds anything of value, because the p-values of the supervised "extracted motion features" (Figure 4) are better than the cluster p-values from the same dataset (Figure 6E).

Response:

The reviewer raises an interesting point. Indeed, when looking exclusively at the p-values, the supervised classifiers are already sufficient to differentiate stress from control mice with high accuracy. However, the strength of the unsupervised analysis of behavior is that potentially novel behavioral features of animals can be detected, that may explain additional variation in the data set that is not covered by pre-annotated behaviors. This is in fact a major strength of the DeepOF package, as it allows for the flexible analysis of rodent behavior in various experimental settings. For example, the DeepOF package can be used to assess sex differences, various disease models, ageing or home cage behavior, to just name a few examples. In these scenarios it is then of great benefit to not only rely on a limited set of pre-defined behavioral features, but also to extract novel and so far unexplored behavioral dimensions that may distinguish the experimental animals better. We have now addressed this point more specifically in the discussion (see "DeepOF as part of a markerless pose estimation toolset", starting from line 713 in page 25).

- More context for the clusters should be provided. Can a human visually recognize some logic behind the clusters when looking at example sequences? Intuitively, one would expect these to correlate to "classical" behaviours as determined with the supervised module, since the underlying data are a mixture of unsupervised and supervised data. However, no such comparison is provided. The authors should investigate how their classical behaviours relate/map to the unsupervised cluster. For example, are multiple behaviours mapped to the same cluster or multiple clusters mapped to the same behaviour? How diverse are sequences that are contained in the same cluster? Are some clusters more likely to happen in certain areas of the behaviour setup? Are some clusters more likely to happen when the two animals are close?

Response:

We agree with the reviewer that the interpretability of the unsupervised analysis and identified behavioral clusters is of key importance and have now further strengthened this aspect in the revised manuscript and DeepOF package. We understand that this part of the question is related to the confusion carried brought from the previous point but would like to clarify once again that no supervised data was used in any step of the unsupervised pipeline, other than post-hoc analysis of the clusters after they were obtained.

Regarding cluster explainability and mapping to classical behaviors, the SHAP analysis picks up positive and negative relationships between the supervised annotations and the clusters detected by the unsupervised pipeline, allowing for a direct comparison. Interestingly, several behavioral clusters from

the unsupervised pipeline can map to the same supervised classifier, as they may detect different aspects or sub-categories of the same behavior. An example is huddling behavior, which can also include freezing behavior, and both behavioral categories are generally affected by chronic stress exposure. Due to the detailed insight provided by the SHAP analysis in combination with the visual verification of the original videos it is then possible to precisely define the individual features of the newly identified behavioral clusters and, if of interest, use them to define new supervised classifiers. Moreover, a key advantage of motion tracking data over other modalities is that one can map all patterns back to their original video sample. DeepOF is fully capable of outputting sample video with just a few commands, as described in the now available tutorials. Furthermore, sample video output for each cluster is now available as supplemental material for single and multi-animal SI, OF, and SA settings.

It must also be highlighted that, while most differentially expressed clusters are, not all clusters remain interpretable by naked eye. In this regard, we highlight the results showcased in figure 7B and in supplemental figures 17B and 18B, which show that a multi-output classifier is capable of detecting very well all different clusters in all settings in unseen data. This strongly suggests that, even when visual inspection is insufficient to label a certain behavior, all clusters are dominated by consistent patterns related by the feature set selected for explanations. This further highlights the complementary nature of visual inspection and post-hoc cluster analysis, where depending on the case each can be more informative than the other.

Regarding the spatial distribution of the expressed clusters, we believe this is a truly insightful comment, even though the usage of round arenas makes it less informative than in other richer settings. We have therefore implemented a function within DeepOF to show the spatial distribution of specific clusters across all animals as heatmaps and use it to show how (in strikingly all cases) CSDS-enriched clusters show occupy the center of the arena less than NS-enriched clusters. It is worth highlighting, here, that DeepOF is agnostic to absolute location in the arena (and it can therefore not use location to assign the clusters in the first place). These analyses were incorporated in figures 6F and supplemental figures 7F and 8F for selected clusters. Heatmaps for all clusters on each setting can be found in supplemental figures 14, 15, and 16.

- The fact that only one animal is tracked in a social interaction test is disappointing, given the advances in multi-animal tracking. It would be interesting to take the behavior/position of the social partner into consideration. This would also make the paper more cutting-edge.

Response:

We fully agree with the reviewer that this was a so-far unexplored, but highly relevant potential of DeepOF and we therefore now included multi animal embeddings within the package.

This is made possible by a fully redesigned embedding pipeline, that takes advantage of graph representation learning to include individual body part speed and distances in a spatially coherent manner. Thus, the version of the software presented in this revised manuscript also introduces multi-animal embeddings. By adding nose-to-nose, nose-to-tail, and tail-to-tail links across all available animals, joint individual behaviors and social interactions on two or more mice are now fully supported. The results presented in the manuscript now show how, while the captured effects of CSDS are comparable across single and multi-animal embeddings, retrieved behaviors are specific to each pipeline. We believe this tool has strong potential to uncover behavioral differences in complex social systems outside CSDS and thank the reviewers for their suggestion.

The pipeline is described in detail in the methods' section (see "Graph representations" starting from line 235 in page 9, and "Unsupervised deep embeddings with DeepOF", starting from line 249 in page

10), and the results obtained with it are discussed throughout the results section (see supplemental figures 8, 15, and 17).

- Presumably the biggest concern is transferability. It is unclear how well DeepOF would perform in a different OFT set-up, with a different mouse strain or in a different lab altogether. The authors state that the ease of use and generalization of the method are its main advantage. However, it needs to first be demonstrated that the provided classifiers are transferable, or if they are subject to strong dataset specific biases. Even small differences in animal age / strain / testing environment can lead to classifiers that struggle to produce reliable data, not to mention mice whose behaviour has been more dramatically altered by an experimental manipulation (e.g. drug treatments etc.). These questions should at the very least be tested on more datasets from the same lab, ideally on datasets from other labs (several can be accessed publicly).

Response:

We fully agree with the reviewer that transferability is a key issue that is a major problem for already available analysis packages and had not been clearly demonstrated in the original manuscript. To address this concern, we now included more details on the internal validation pipeline for the pre-trained models, as well as on the datasets employed.

Moreover, we have extended our validation pipeline with an additional dataset coming from a different group, using different strains and in a different arena setup. We clearly show that DeepOF performs comparably in this environment than in videos obtained in our lab. Thus, an external validation and cross-validation across four different data sets obtained in four different settings and in different laboratories. Importantly, the performance of DeepOF is stable across these data sets and the corresponding data are now included as **supplemental figure 2A**.

- Related to the previous point, could the algorithm be applied to the social avoidance test? This test is conducted in a similar size arena (albeit a square rather than a circle). Since most labs use square open field tests, this would already test some of the issues surrounding transferability.

Response:

We believe this is an excellent suggestion, which was also raised by another reviewer and which we now incorporated in two orthogonal parts of the revised version.

First, the social avoidance data was used as part of the testing pipeline for the huddling classifier, as depicted in supplemental figure 2A. The classifier shows comparable high performance on the social avoidance videos even when no data from this setting is available, suggesting good transferability as pointed out by the reviewer.

Furthermore, we ran a full unsupervised analysis with DeepOF on the social avoidance data, to characterize the potential behavioral differences in both trials (Trial 1; without social conspecific and Trial 2; with social conspecific). Interestingly, no different behavioral patterns were observed for the social avoidance trial 1 data between the non-stressed (NS) and stressed (CSDS) animals, however clear differences emerged in the social avoidance trial 2 data (see new supplemental figure 9). The data indicate that in addition to this validated measure of social avoidance, DeepOF enables the detection of additional behavioral alterations in the experimental animals, some of which might be even unrelated to the social avoidance per se and rather reflect additional behavioral disturbances induced by chronic stress exposure.

Taken together, these results show how both pipelines offered by DeepOF can be applied to datasets using experimental settings other than the round open fields used in the main sections of this study. Moreover, they provide specific insights into the requested social avoidance task.

- I did not find any online repository of the data. Neither video recording, nor pose-estimation data are accessible. It is necessary to deposit the raw video recordings for other labs to establish the approach using the same data, before testing it on their own data.

Response:

We agree with the reviewer that depositing the data and raw video recordings for other users is a great idea and have acted in this regard.

A password protected online shared folder was consequently set up in the Max Planck DataShare services, which allowed us to host raw videos, the code used to generate the figures (as jupyter notebooks), DeepLabCut tracks, DeepOF projects, outputs (including representative snippets) and trained models, and most labelled data used to validate and test the supervised models. A readme file is included in the home directory with a complete description of how the repository is organized.

There are only two current restrictions:

- The now privately shared raw videos for the SI, SA, and OF datasets, as well as the included labelled behaviors were shared with the current edition of the MaBe challenge. Until the challenge is finished (July 2023) we cannot share the data publicly.
- The raw videos, tracks, and labelled frames from the EX dataset are currently being used in another project, and our collaborators cannot disclose the full dataset for the time being. In a spirit of transparency, they however provided sample videos and tracking files, which are uploaded to the shared folder.

We are confident the data and code mentioned here, together with the open source nature of DeepOF, will help guarantee the reproducibility of the presented results.

Minor comments:

- Throughout the results section a clearer distinction is needed between the OFT test (where the animal is tested alone: OFT-alone), and the OFT test where a second mouse was introduced (social interaction test). It was often unclear, which results pertain to which “test”. This starts with a clearer visualization in Figure 2 when which test was run and which analyses in the subsequent figures relate to which test.

Response:

We thank the reviewer for the feedback on the confusing different figures and have now included a graphical representation of the different tasks in Figure 2A. Also, the names of the tests have been added in the different figures.

- Lines 342-344: The rationale for analysing time bins (and how those bins were chosen) is unclear. If time bins were analysed, why wasn't this done for the OFT-alone and the Social-Interaction datasets?

Response:

We thank the reviewer for pointing to this discrepancy in our analyses. We have now included the full time bin analysis, including PCA distinctions for the open field data as well (see new supplemental

figure 3). We have also added the reasoning for the 2.5 minute bins in the results section. Further, we have also greatly expanded in the time-bin rationale for the unsupervised analysis in the methods' section (see "Time binning and habituation quantification", starting from line 326 in page 12).

- Why was the social defeat stress run for 21 days, but behaviour only analysed after day 15? Were no behaviour tests conducted on day 21? Figure 2A suggests differently. This would be an important opportunity to test how reproducible the data are or how variable behaviour might still be between D15 and D21. It's a bit strange to correlate behaviour with many physiological measures taken a week later.

Response:

To validate the utility of the DeepOF toolbox we applied the 21-day version of the CSDS protocol previously utilized by us and others (Haenisch et al., 2009; Kudryavtseva et al., 1991; Wagner et al., 2012). A main advantage of this protocol is that the CSDS phenotype is induced in the first 2 weeks and can then be measured in subsequent behavioral and physiological tests while the stress continues in week 3. This enables multiple readouts over several days without needing to account for variable recovery periods after the end of the stress procedure. Indeed, the 3-week CSDS protocol therefore also allows for repeated testing and measures of behavioral stability over time. Further, the prolonged 3-week CSDS paradigm was specifically chosen to elicit a more profound passive defeat phenotype, as originally reported by (Kudryavtseva et al., 1991), now clarified in the methods (**page 14, lines 373-376**). To clarify the experimental timeline, the corresponding figure has been updated, to include directly the days of the specific behavioral tests (Figure 2A). In addition, as the reviewer points out correctly, a downside of this approach is that behavioral readouts and physiological markers obtained at sacrifice differ substantially in time and we now mention this limitation in the text (**page 20, lines 549-551**). However, this methodological detail does not impact our conclusions on the validity of the DeepOF toolbox in quantifying individual and social behavioral profiles.

- In figure 5C, is the strongest correlation found for sniffing should be discussed. What might this mean? Is this sniffing against the wall as depicted in Figure 1C? An exploratory behaviour?

Response:

The reviewer raises an interesting point. The individual behaviors that were found to significantly correlate with the Z-score of SI, like speed, distance, inner zone entries, and look-around in the open field arena, are likely related to aspects of social anxiety. We now mention this interpretation in the results section (page 20, lines 557-560). However, the wealth of behavioral classifiers obtained from both supervised and unsupervised behavioral analyses are still in stark contrast to our understanding of the motivational drive of individual behaviors and further research will be necessary to allow a better understanding and interpretation of the individual behavioral features.

- Line 429: entropy is an interesting calculation. It's not clear, however, how this helps to "explain" animal behaviour. Does entropy in the social interaction data correlate with certain behaviours? Or with the social avoidance score? Or with the physiological stress score?

Response:

After careful thought, we agree that entropy, in this case, may not be the most informative or readily interpretable measure, and we have decided to remove it from our analysis in favor of the heatmaps indicating the spatial expression of clusters along the arena, also suggested by the reviewer. However, we would like to point out that the previously provided analyses and plots can still be conducted/obtained within DeepOF for those who are interested.

Literature

- Golden, S. A., Covington, H. E., Berton, O., & Russo, S. J. (2011). A standardized protocol for repeated social defeat stress in mice. *Nature Protocols*, 6(8), 1183–1191. <https://doi.org/10.1038/nprot.2011.361>
- Haenisch, B., Bilkei-Gorzo, A., Caron, M. G., & Bönisch, H. (2009). Knockout of the norepinephrine transporter and pharmacologically diverse antidepressants prevent behavioral and brain neurotrophin alterations in two chronic stress models of depression. *Journal of Neurochemistry*, 111(2), 403–416. <https://doi.org/10.1111/j.1471-4159.2009.06345.x>
- Kleinberg, J. (2002). Bursty and hierarchical structure in streams. *Proceedings of the Eighth ACM SIGKDD International Conference on Knowledge Discovery and Data Mining*, 91–101. <https://doi.org/10.1145/775047.775061>
- Kudryavtseva, N. N., Bakshtanovskaya, I. V., & Koryakina, L. A. (1991). Social model of depression in mice of C57BL/6J strain. *Pharmacology Biochemistry and Behavior*, 38(2), 315–320. [https://doi.org/10.1016/0091-3057\(91\)90284-9](https://doi.org/10.1016/0091-3057(91)90284-9)
- Labonté, B., Jeong, Y. H., Parise, E., Issler, O., Fatma, M., Engmann, O., Cho, K.-A., Neve, R., Nestler, E. J., & Koo, J. W. (2019). Gadd45b mediates depressive-like role through DNA demethylation. *Scientific Reports*, 9(1), 4615. <https://doi.org/10.1038/s41598-019-40844-8>
- Lorsch, Z. S., Ambesi-Impiombato, A., Zenowich, R., Morganstern, I., Leahy, E., Bansal, M., Nestler, E. J., & Hanania, T. (2021). Computational Analysis of Multidimensional Behavioral Alterations After Chronic Social Defeat Stress. *Biological Psychiatry*, 89(9), 920–928. <https://doi.org/10.1016/j.biopsych.2020.10.010>
- Luxem, K., Mocellin, P., Fuhrmann, F., Kürsch, J., Miller, S. R., Palop, J. J., Remy, S., & Bauer, P. (2022). Identifying behavioral structure from deep variational embeddings of animal motion. *Communications Biology*, 5(1), 1267. <https://doi.org/10.1038/s42003-022-04080-7>
- Morel, C., Montgomery, S. E., Li, L., Durand-de Cuttoli, R., Teichman, E. M., Juarez, B., Tzavaras, N., Ku, S. M., Flanigan, M. E., Cai, M., Walsh, J. J., Russo, S. J., Nestler, E. J., Calipari, E. S., Friedman, A. K., & Han, M.-H. (2022). Midbrain projection to the basolateral amygdala encodes anxiety-like but not depression-like behaviors. *Nature Communications*, 13(1), 1532. <https://doi.org/10.1038/s41467-022-29155-1>
- Nilsson, S. R. O., Goodwin, N. L., Choong, J. J., Hwang, S., Wright, H. R., Norville, Z. C., Tong, X., Lin, D., Bentzley, B. S., Eshel, N., McLaughlin, R. J., & Golden, S. A. (2020). Simple Behavioral Analysis (SimBA) – an open source toolkit for computer classification of complex social behaviors in experimental animals. *BioRxiv*, 2020.04.19.049452. <https://doi.org/10.1101/2020.04.19.049452>
- Pereira, T. D., Tabris, N., Matsliah, A., Turner, D. M., Li, J., Ravindranath, S., Papadoyannis, E. S., Normand, E., Deutsch, D. S., Wang, Z. Y., McKenzie-Smith, G. C., Mitelut, C. C., Castro, M. D.,

D'Uva, J., Kislin, M., Sanes, D. H., Kocher, S. D., Wang, S. S.-H., Falkner, A. L., ... Murthy, M. (2022). SLEAP: A deep learning system for multi-animal pose tracking. *Nature Methods*, *19*(4), 486–495. <https://doi.org/10.1038/s41592-022-01426-1>

Segalin, C., Williams, J., Karigo, T., Hui, M., Zelikowsky, M., Sun, J. J., Perona, P., Anderson, D. J., & Kennedy, A. (2021). The Mouse Action Recognition System (MARS) software pipeline for automated analysis of social behaviors in mice. *ELife*, *10*. <https://doi.org/10.7554/eLife.63720>

Torres-Berrío, A., Nouel, D., Cuesta, S., Parise, E. M., Restrepo-Lozano, J. M., Larochelle, P., Nestler, E. J., & Flores, C. (2020). MiR-218: a molecular switch and potential biomarker of susceptibility to stress. *Molecular Psychiatry*, *25*(5), 951–964. <https://doi.org/10.1038/s41380-019-0421-5>

Wagner, K. V., Marinescu, D., Hartmann, J., Wang, X.-D., Labermaier, C., Scharf, S. H., Liebl, C., Uhr, M., Holsboer, F., Müller, M. B., & Schmidt, M. V. (2012). Differences in FKBP51 Regulation Following Chronic Social Defeat Stress Correlate with Individual Stress Sensitivity: Influence of Paroxetine Treatment. *Neuropsychopharmacology*, *37*(13), 2797–2808. <https://doi.org/10.1038/npp.2012.150>

REVIEWER COMMENTS

Reviewer #1 (Remarks to the Author):

No further comments

Reviewer #2 (Remarks to the Author):

I thank the authors for their thorough and careful revision of the manuscript. They have satisfactorily addressed all of my comments with the addition of extra readouts, methods and discussion.

Reviewer #3 (Remarks to the Author):

While the reviewers' comments were challenging, the authors have provided a very extensive rebuttal and added a lot of important information and new analyses. The fact that DeepOF now assesses the movement and interaction of two animals is a major improvement and makes the tool much more state-of-the-art than before. It's also laudable that appropriate multiple testing corrections were applied throughout the manuscript, particularly when comparing multiple clustering results. Considering these improvements, and the option for users to select either supervised or unsupervised analyses, the paper represents a nice addition to the rapidly expanding toolbox for analyzing animal behavior.

Upon carefully re-reading the manuscript, a few additional comments arose, which should still be taken into consideration:

- 1) The term "huddling" is problematic; it slipped my attention during the initial review. Huddling is what mice do in groups: <https://mousebehavior.org/group-sleeping-huddling/> . The authors' description does not fit that of huddling (and apparently the animal does it alone). Do the authors mean grooming-related behaviors? It is important to specify and correct the wording, as this will lead to confusion otherwise. Huddling should only be used in a social context.
- 2) Please report the lighting conditions in the room and the day-light-cycle of the animal facility
- 3) Figure 2B: the y-axis must start at zero
- 4) Suppl Figure 4 should replace Main Figure 4 (there is no space constraint, it carries important information, Fig 4 is very small anyway).
- 5) Figure 5 is a bit thin and simple, the z-score correlation provides proof-of-principle, but drawing conclusions about stress-responsiveness of individual mice is not demonstrated convincingly. It's also questionable whether this is even possible given the innate variance in animal behavior and the single OF measurement. In this context, Supplementary Figure 6 is very interesting and should be added to main Figure 5. These results shouldn't be buried in the supplementary text, especially given that the current main figures leave plenty of room for more data.
- 6) Line 605: should say "supplemental" figure 8D

7) While displaying the spatial distribution of the clusters (e.g. Fig.6F) is potentially useful, the interpretation that clusters enriched in control animals occur throughout the arena and CSDS-enriched clusters occur closer to the walls is not substantiated by the data. Fig. 6F suggests that also cluster 4 – more prevalent in control mice – occurs predominantly in the periphery and hardly ever in the center. This also raises the question of how much information is actually gained by this analysis, if it remains unclear which behaviors the clusters actually represent (likely a mix of many behaviors, and all behaviors are biased towards the periphery). Clusters 4 and 5 seem to be the only interesting exception to this in the OF task (Suppl Figure 16), these clusters are less frequent in CSDS mice (Suppl Fig 8E). It would be worth to report what these clusters represent (maybe just center time?), and to tone down the statements about spatial distribution of clusters.

8) The authors should provide video examples of the clusters, as commonly done in this field (e.g. MoSeq papers). Particularly the description on page 24 would be aided by seeing the actual behaviors.

9) Now that the manuscript focuses much more on the social interaction (with 2-animal tracking), it comes a lot closer to SIMBA, which has been a widely used open-source tool for social behavior analysis for 3 years now. I would find it fair to explore the option of a back-to-back publication of both papers.

REVIEWER COMMENTS

Reviewer

#1

No further comments

Reviewer

#2

I thank the authors for their thorough and careful revision of the manuscript. They have satisfactorily addressed all of my comments with the addition of extra readouts, methods and discussion.

Reviewer

#3

While the reviewers' comments were challenging, the authors have provided a very extensive rebuttal and added a lot of important information and new analyses. The fact that DeepOF now assesses the movement and interaction of two animals is a major improvement and makes the tool much more state-of-the-art than before. It's also laudable that appropriate multiple testing corrections were applied throughout the manuscript, particularly when comparing multiple clustering results. Considering these improvements, and the option for users to select either supervised or unsupervised analyses, the paper represents a nice addition to the rapidly expanding toolbox for analyzing animal behavior.

Upon carefully re-reading the manuscript, a few additional comments arose, which should still be taken into consideration:

1. The term "huddling" is problematic; it slipped my attention during the initial review. Huddling is what mice do in groups: <https://mousebehavior.org/group-sleeping-huddling/>. The authors' description does not fit that of huddling (and apparently the animal does it alone). Do the authors mean grooming-related behaviors? It is important to specify and correct the wording, as this will lead to confusion otherwise. Huddling should only be used in a social context.

Response:

We thank the reviewer for making this critical point and agree that the terminology for every behavior needs to be weighed carefully. The confusion seems to have arisen from the adoption of the term "huddling" in our manuscript as an adaptation of the "stopped and huddled" trait described by Chaumont et al. in their 2019 paper "Real-time analysis of the behaviour of groups of mice via a depthsensing camera and machine learning". We agree with the reviewer that the term 'huddling' is confusing, and changed it to 'stopped and huddled' as in the cited paper throughout the manuscript. A clarification has been added to the legend in figure 1, and code has been adapted in DeepOF to label the behavior "huddle" (as an abbreviation of "stopped-and-huddled") instead of "huddling".

Literature

De Chaumont, F., Ey, E., Torquet, N., Lagache, T., Dallongeville, S., Imbert, A., ... & Olivo-Marin, J. C. (2019). Real-time analysis of the behaviour of groups of mice via a depth-sensing camera and machine learning. *Nature biomedical engineering*, 3(11), 930-942.

2. Please report the lighting conditions in the room and the day-light-cycle of the animal facility.

Response:

The information regarding the animal housing conditions and day-light-cycle can now be found at lines 362-366.

3. Figure 2B: the y-axis must start at zero

Response:

We have now revised the graph so that the y-axis starts at 0. An indent has been made, however, for the graph to depict the normal distribution of mouse body weights at 2-3 months of age, which is in the range of 26-30 grams, and to increase readability (see revised figure 1).

4. Suppl Figure 4 should replace Main Figure 4 (there is no space constraint, it carries important information, Fig 4 is very small anyway).

Response:

We agree with the reviewer and have implemented this change.

5. Figure 5 is a bit thin and simple, the z-score correlation provides proof-of-principle, but drawing conclusions about stress-responsiveness of individual mice is not demonstrated convincingly. It's also questionable whether this is even possible given the innate variance in animal behavior and the single OF measurement. In this context, Supplementary Figure 6 is very interesting and should be added to main Figure 5. These results shouldn't be buried in the supplementary text, especially given that the current main figures leave plenty of room for more data.

Response:

We agree with the reviewer and have now added the data of former Supplementary Figure 6 to the main Figure 5.

6. Line 605: should say "supplemental" figure 8D

Response:

We thank the reviewer for pointing at this mistake and have corrected the text accordingly.

7. While displaying the spatial distribution of the clusters (e.g. Fig.6F) is potentially useful, the interpretation that clusters enriched in control animals occur throughout the arena and CSDS-enriched clusters occur closer to the walls is not substantiated by the data. Fig. 6F suggests that also cluster 4 – more prevalent in control mice – occurs predominantly in the periphery and hardly ever in the center. This also raises the question of how much information is actually gained by this analysis, if it remains unclear which behaviors the clusters actually represent (likely a mix of many behaviors, and all behaviors are biased towards the periphery). Clusters 4 and 5 seem to be the only interesting exception to this in the OF task (Suppl Figure 16), these clusters are less frequent in CSDS mice (Suppl Fig 8E). It would be worth to report what these clusters represent (maybe just center time?), and to tone down the statements about spatial distribution of clusters.

Response:

Upon careful review, we agree with the statement made about the spatial distribution of cluster expression: while potentially interesting in more complex arenas (to check if behaviors are expressed closer to certain spatial cues, for example) the overall bias towards the periphery in our setting seems prevalent throughout clusters. We have revised both figures and text to clarify this issue (see lines 650-666 in the main text, as well as figure legends). Moreover, we have removed the NS and CSDS enriched cluster heatmaps from figures 6 and supplemental 5 and 6 (although they are still accessible in supplemental figures 12, 13, and 14), and left the overall locomotion heatmap as a representative example. We also reintroduced the behavioral entropy score left out in the previous version of the manuscript as a replacement and furthered the analysis to show differences in entropy in both single and multi-animal embeddings in the SI setting, that significantly correlate with stress physiology (as suggested by the reviewer in the first revision). We also show that, in accordance with the rest of the paper, such differences are not detectable in the open field setting. We believe the current state of the figures referred to in this comment is such that all panels contribute relevant information to the message of the paper.

8. The authors should provide video examples of the clusters, as commonly done in this field (e.g. MoSeq papers). Particularly the description on page 24 would be aided by seeing the actual behaviors.

Response:

We thank the reviewer for this pertinent comment. We would like to clarify that video examples are indeed available as part of the supplementary material provided (see code and data availability statement). The MPCDF-hosted shared folder linked there contains a README file indicating the exact location of the cluster videos.

9. Now that the manuscript focuses much more on the social interaction (with 2-animal tracking), it comes a lot closer to SIMBA, which has been a widely used open-source tool for social behavior analysis for 3 years now. I would find it fair to explore the option of a back-to-back publication of both papers.

Response:

We thank the reviewer for the suggestion. We agree that publishing back-to-back with a SimBA-related manuscript would be ideal, if indeed such a manuscript was submitted to Nature Communications and is accepted for publication. We would only request that this does not lead to significant delays in the publication process.

REVIEWERS' COMMENTS

Reviewer #3 (Remarks to the Author):

The authors have thoroughly addressed all my comments.

NCOMMS-22-37587B – rebuttal letter

REVIEWER COMMENTS

Reviewer #3

The authors have thoroughly addressed all my comments.

Response: We thank the reviewer for his/her positive evaluation of our revised manuscript.